# Metamizer: a versatile neural optimizer for fast and accurate physics simulations

**Nils Wandel, Stefan Schulz & Reinhard Klein**
Department of Computer Science
University of Bonn
53115 Bonn, Germany
`wandeln@cs.uni-bonn.de`

## Abstract

Efficient physics simulations are essential for numerous applications, ranging from realistic cloth animations in video games, to analyzing pollutant dispersion in environmental sciences, to calculating vehicle drag coefficients in engineering applications. Unfortunately, analytical solutions to the underlying physical equations are rarely available, and numerical solutions are computationally demanding. Latest developments in the field of physics-based Deep Learning have led to promising efficiency gains but still suffer from limited generalization capabilities across multiple different PDEs.

Thus, in this work, we introduce Metamizer, a novel neural optimizer that iteratively solves a wide range of physical systems without retraining by minimizing a physics-based loss function. To this end, our approach leverages a scale-invariant architecture that enhances gradient descent updates to accelerate convergence. Since the neural network itself acts as an optimizer, training this neural optimizer falls into the category of meta-optimization approaches. We demonstrate that Metamizer achieves high accuracy across multiple PDEs after training on the Laplace, advection-diffusion and incompressible Navier-Stokes equation as well as on cloth simulations. Remarkably, the model also generalizes to PDEs that were not covered during training such as the Poisson, wave and Burgers equation.

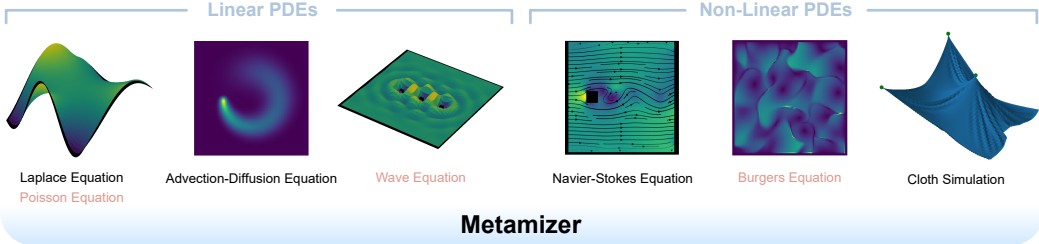

Figure 1: Metamizer is able to simulate various linear and non-linear physical systems. All of the depicted results were produced by the same neural model (same architecture and same weights). PDEs marked in red were not considered during training.

## 1 Introduction

Countless physical systems can be described by partial differential equations (PDEs). For example, electrostatic or gravitational fields can be described by the Poisson equation, heat diffusion or pollutant dispersion can be described by the advection-diffusion equation, pressure waves by the wave equation, fluids by the incompressible Navier-Stokes equation, and so on. Unfortunately, most of these equations do not have analytical solutions, so numerical solvers must be used. To reduce the computational burden of numerical methods, numerous neural surrogate models have been developed in recent years that achieve fast and highly efficient physics simulations (Thuerey et al., 2021;

Cuomo et al., 2022). Moreover, recent developments (Kaneda et al., 2023; Chen et al., 2024) show that neural networks do not fall behind numerical solvers in terms of accuracy anymore. However, neural surrogate models are usually tailored to specific PDEs and do not generalize well across multiple different PDEs without retraining. Zero-shot PDE solvers, analogous to current endeavours in foundational models for natural language processing (Brown, 2020) or computer vision (Radford et al., 2021; Kirillov et al., 2023), could be of great benefit for instant realisitc animations in computer games or computer aided engineering.

Thus, in this work, we propose Metamizer, a neural optimizer to accelerate gradient descent for fast and accurate physics simulations. To this end, we introduce a novel *scale-invariant architecture* that suggests improved gradient descent update steps that are independent of arbitrary scalings of the loss function or loss domain. We train Metamizer *without any training data* directly on the physics-based loss that it is supposed to optimize. After training, unlike traditional gradient descent methods, Metamizer requires *no tuning* of parameters such as learning rate or momentum, and allows *tradeoffs between runtime and accuracy* for *fast and highly accurate* results. We evaluate the very same network (same architecture and same weights) across a wide range of linear and non-linear PDEs and showcase its *generalization capabilites - even to PDEs that were not covered during training*.

Code as well as a pretrained model are available on Github.

## 2 RELATED WORK

**Numerical Methods** typically rely on a discretization scheme (e.g. Finite Differences, Finite Volumes, Finite Elements etc. (Peiró & Sherwin, 2005)) in order to transform PDEs into large systems of equations. Depending on properties such as symmetry or linearity of this system, various numerical solvers such as (nonlinear) CG, GMRES, MINRES etc. can be employed. To alleviate their computational burden (Farmaga et al., 2011), already in the last century, several specialized methods have been developed to accelerate for example fluid simulations (Chen & Doolen, 1998; Stam, 1999) or cloth simulations (Baraff & Witkin, 1998). More recently, neural preconditioners (Kaneda et al., 2023; Lan et al., 2023) significantly accelerated conjugate gradient methods for the Poisson equation. However, these specialized methods exploit domain-specific assumptions and, thus, do not generalize well across different linear and nonlinear PDEs.

**Gradient Descent** based methods like AdaGrad (Duchi et al., 2011), Adam (Kingma & Ba, 2017; Reddi et al., 2019), AdamW (Loshchilov, 2017) and many more (Norouzi & Ebrahimi, 2019) build the backbone of most Deep Learning approaches. These optimization methods can be generally applied to a wide range of differentiable minimization problems including solving PDEs (Nurbekyan et al., 2023). However, Gradient Descent methods require tuning of hyperparameters such as learning rate or momentum and many iterations until convergence. As a result, they are not yet efficient enough for real-time physics simulations.

**Meta-Learning,** and in particular "Learning to Optimize" (L2O), is the field of research that deals with the automatic improvement of optimization algorithms, typically by means of machine learning techniques. Pytorch libraries such as "higher" (Grefenstette et al., 2019) or "learn2learn" (Arnold et al., 2020) allow for example to automatically optimize hyperparameters of optimizers like Adam through gradient descent. Andrychowicz et al. (2016) train LSTMs to optimize neural networks for classification tasks and style transfer and Chen et al. (2020) present several training techniques that suggest there is still room for potential improvement in the field L2O. However, to the best of our knowledge, L2O has not yet been applied in the context of physics simulations.

**Data-driven Deep Learning approaches** have been widely established to generate neural surrogate models for efficient physics simulations for example in context of fluid dynamics (Tompson et al., 2017; Sanchez-Gonzalez et al., 2020), cloth simulations (Pfaff et al., 2020), weather forecasting (Bonev et al., 2023; Lam et al., 2023) or aerodynamics (Li et al., 2024). While these methods allow efficient simulations at coarse spatial and temporal resolutions or on physical systems without knowledge of the underlying mathematical equations, they require large amounts of high-quality training data, which can be expensive to generate. Furthermore, generalization and accuracy beyond the training data is often fairly limited.

**Physics-driven Deep Learning approaches** in contrast, require only little or no ground truth data and rely on physics-based loss. Here, 2 major strands of research can be discerned:

First, implicit neural representations (often referred to as *PINNs*) as popularized by Raissi et al. (2019) have become a new and quickly growing field of research (Cuomo et al., 2022) ranging from climate simulations (Chen et al., 2022; Lai et al., 2024) to highly accurate simulations of Burgers equation (Chen et al., 2024) and fluid dynamics (Cai et al., 2021; Ghosh et al., 2023), all the way to robotics (Sanyal & Roy, 2023; Liu et al., 2024). Unfortunately, the learned implicit representations need to be retrained for every physical simulation and thus are not real-time capable.
Second, neural surrogate models that evolve an explicit state representation in time as presented by Zhu et al. (2019) have been successfully applied to the coupled 2D Burgers equation (Geneva & Zabaras, 2020), incompressible fluid dynamics in 2D and 3D (Wandel et al., 2021a;b) and cloth simulations (Santesteban et al., 2022; Stotko et al., 2024). While these methods do not require any training data and show good generalization performance within their PDE domain, so far, they do not generalize across multiple PDEs.

## 3 FOUNDATIONS

In this chapter, we introduce some basic concepts and definitions for PDEs, explain, how finite difference schemes can be used to formulate a physics-based loss, and give some intuition to motivate our gradient-based optimization approach.

**Partial Differential Equations (PDEs)** Partial differential equations are equations that constrain the partial derivatives of a multivariate function. In its most general form, a PDE can be written as:

$$F(x_1, ..., x_n, u, \partial_{x_1} u, ..., \partial_{x_n} u, \partial_{x_1}^2 u, ...) = 0 \,\forall (x_1, ..., x_n) \in \Omega \qquad (1)$$

where $u(x_1, ..., x_n)$ is the *unknown function* that we want to solve, $x_1, ..., x_n$ are the *independent variables* inside a domain $\Omega$, and $\partial_{x_i} u$ denotes the partial derivative of $u$ with respect to $x_i$. If one of the independent variables corresponds to time $t$, $F$ is called a *time-dependent* PDE. If $u$ does not depend on $t$, it is called *stationary*. If $F$ is linear in $u$ and its derivatives, $F$ is called a *linear* PDE. Otherwise, $F$ is called a *non-linear* PDE.

**Boundary and Initial Conditions** Usually, additional constrains at the domain boundaries are given such as boundary or initial conditions. Although various types of different boundary conditions could be considered, in this work, for simplicity, we focus solely on *Dirichlet boundary conditions*:

$$u(x_1, ..., x_n) = d(x_1, ..., x_n) \,\forall (x_1, ..., x_n) \in \partial\Omega \qquad (2)$$

Here, $d(x_1, ..., x_n)$ directly specifies the values of $u$ at the domain boundary $\partial\Omega$. Typically, time-dependent PDEs also require an initial state (or *initial condition*) from which the solution of a PDE evolves over time.

**Finite Differences** To compute $u$ and its partial derivatives in a numerical manner, $u$ needs to be discretized. Here, we consider a regular grid in space (and time). This allows to compute derivatives like gradients ($\nabla$), divergence ($\nabla\cdot$), Laplace ($\Delta$) or curl ($\nabla\times$) numerically by performing convolutions with finite difference kernels. By plugging the results of these finite difference convolutions for every grid point into the general formulation of PDEs (Equation 1), we obtain a large system of equations. If the PDE is linear, this set of equations becomes a (typically sparse) linear system of equations.

**Time-Integration Schemes** In case of time-dependent PDEs, a strategy must be chosen to integrate the simulation in time for given initial conditions. There exist several strategies to calculate and integrate the time derivative with finite differences. The most popular strategies are:

- *Forward Euler:* Here, the computation of the time derivative depends only on the current time step. This explicit strategy makes computations easy and efficient, but tends to cause unstable simulation behavior for many physical systems if the time step is chosen too large. Therefore, explicit schemes are not considered in this work.

- *Backward Euler:* Here, the computation of the time derivative depends on the following time step. This implicit strategy results in stable simulations, but can suffer from numerical dissipation and requires solving large systems of equations.

- *Crank-Nicolson:* Here, the centered finite difference of the current and the following time step is used to calculate the time derivative. This implicit strategy results in more accurate computations compared to forward or backward Euler, but requires, like backward Euler, solving large systems of equations.

**Physics-Based Loss**  Solving large (potentially non-linear) systems of equations obtained by finite differences is a challenging task that is typically computationally expensive. Thus, we formulate a physics-constrained loss, that penalizes the mean squared residuals (left-hand side of Equation 1) as follows:

$$L(u) = \frac{1}{|\Omega|} \sum_{(x_1,...,x_n) \in \Omega} |F(x_1, ..., x_n, u, \partial_{x_1} u, ..., \partial_{x_n} u, \partial_{x_1}^2 u, ...)|^2 \tag{3}$$

This way, we can search for a solution $u^*$ of the PDE by minimizing $L$:

$$u^* = \arg\min_u L$$

Dirichlet Boundary conditions (Equation 2) can be typically applied directly by setting the corresponding boundary grid points of $u$ equal to $d$. However, sometimes, this is not possible and we have to consider an additional loss term for the Dirichlet boundary conditions:

$$L_D(u) = \frac{1}{|\partial\Omega|} \sum_{(x_1,...,x_n) \in \partial\Omega} |u(x_1, ..., x_n) - d(x_1, ..., x_n)|^2 \tag{4}$$

In this case, we have to minimize $L + L_D$ to solve Equation 1 and Equation 2.

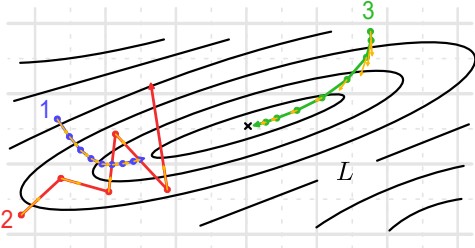

Figure 2: Exemplary visualization of a 2D loss function with a steep valley. Naive gradient descent fails since it either converges very slowly (blue curve 1) or diverges (red curve 2). Ideally, we would like to adapt the step size and direction for accelerated convergence (green curve 3).

**Gradient Descent Intuition**  In order to minimize the physics-based loss $L$ (Equation 3 and 4), we follow a gradient descent approach. Figure 2 provides some intuition on common problems with naive gradient descent in the vicinity of ill conditioned minima and how convergence could be improved: If we take too small steps along the gradients (blue curve 1), convergence becomes very slow. If we take too large steps (red curve 2), the gradient descent scheme diverges. Ideally (green curve 3), we would first take some small initial steps to check the local gradients and avoid initial divergence. Then, the step size can be carefully increased. Instead of directly following the gradients, a better step direction can be found by taking the gradients of the previous steps into account as well. Once we come close to the minimum, the step size should be decreased again to achieve higher accuracy. As discussed by Holl et al. (2022), scaling the loss function $L$ and thus its gradients by a constant factor $c$ does not affect its minima ($\arg\min_u L(u) = \arg\min_u c \cdot L(u)$). Thus, the gradient descent procedure should be invariant with respect to constant gradient scalings. Similarly, scaling the coordinate system of $L$ should result in equally scaled update steps ($\arg\min_u L(u) = c \cdot \arg\min_u L(c \cdot u)$, see solid vs dashed lines in Figure 2). This motivates our scale invariant Metamizer architecture that can deal with arbitrarily scaled gradients and can automatically adapt its step size (see Figure 3 a).

## 4 METHOD

Our approach aims to accelerate convergence of a gradient descent scheme by proposing improved update steps with a neural network.

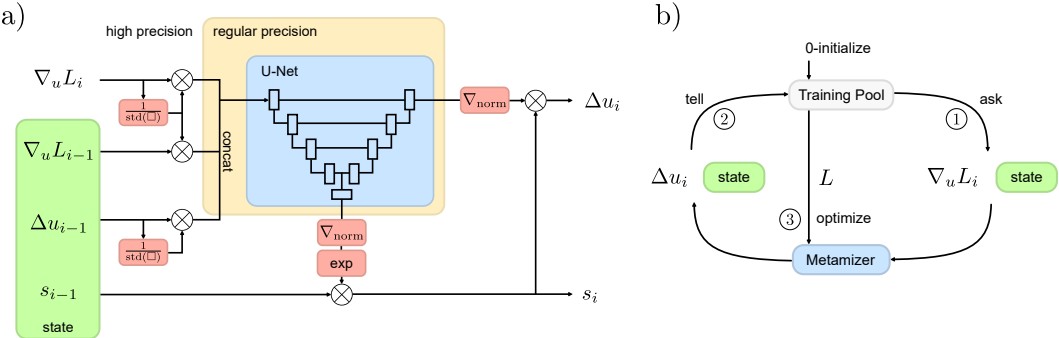

Figure 3: a) Metamizer architecture: a scale invariant neural optimizer. b) Training Cycle.

**Metamizer Architecture** The neural network architecture as depicted in Figure 3 a) works as follows: First, the gradient $\nabla_u L_i$ of the current iteration $i$ as well as the gradient $\nabla_u L_{i-1}$ and update step $\Delta u_{i-1}$ of the previous iteration $i-1$ are taken as input. Then, the gradients of the current and previous iteration are normalized by the standard deviation of the current gradient and the step size is normalized by its standard deviation as well. These values are then concatenated and fed into a U-Net (Ronneberger et al., 2015) in order to predict an update scale factor for $s_i$ and an update direction that gets multiplied with $s_i$ to obtain the update step $\Delta u_i$ for the next iteration. The U-Net architecture was chosen since it allows to model long-range dependencies at coarse resolutions while preserving small details at fine resolutions, reflecting the various domains of dependence of different PDEs. Finally, the computed update step is applied to $u$ to obtain $u_{i+1} = u_i + \Delta u_i$. For the initial state, we set $u_0 = \nabla_u L_0 = \Delta u_0 = 0$ and set $s_0 = 0.05$ to avoid initial divergence by making only small steps at the beginning.

These normalization and scaling measures make the network invariant with respect to arbitrary scalings of the loss or domain and allow the network to make reasonable progress at all stages of the optimization: At the beginning, when large update steps must be made in the presence of large gradients, as well as in later optimization stages, when fine adjustments must be made in the presence of small gradients. To allow for very high accuracies, the gradient computations and update steps need to be calculated in double precision. Nevertheless, due to the input normalizations, the U-Net can be evaluated using a more efficient regular float precision. Since the outputs of the network get scaled by $s_i$, the training gradients get scaled as well. This would result in huge gradients for the first optimization iterations compared to basically no training signal at later optimization iterations. Thus, we employ gradient normalization (Chen et al., 2018) to normalize gradients during backpropagation and provide the network with an evenly distributed training signal during all optimization stages and across all different PDEs that we train on in parallel. Since we want to reuse the same network for different PDEs that might also contain different numbers of channels, Metamizer optimizes all channels separately (e.g. the Laplace equation requires only 1 channel while incompressible fluids require 2 channels for the velocity and pressure field and cloth dynamics require 3 channels to describe motions in $x/y/z$ directions).

**Training** The neural optimizer was trained in a cyclic Meta-learning manner inspired by Wandel et al. (2021a): As visualized in Figure 3 b), we first 0-initialize a training pool of randomized PDEs (e.g. randomized boundary conditions for the Laplace / advection-diffusion / Navier-Stokes equations). Note that in contrast to data-driven methods, no pre-generated ground truth data is needed. Then, we draw a random mini-batch that contains the current gradient $\nabla_u L_i$ of the physics-based loss as well as optimizer state information (i.e. $\nabla_u L_{i-1}, \Delta u_{i-1}, s_{i-1}$) and feed it into the Metamizer architecture to compute an update step $\Delta u_i$. This update step, together with updated state information (i.e. $\nabla_u L_i, \Delta u_i, s_i$) is fed back into the training pool to update the training environments with more realistic data and to recompute the physics-based loss $L$ based on the updated values of $u$. This recomputed loss is then used to optimize the Metamizer with Adam (lr=0.001). By iterating this training cycle, the training pool becomes filled with more and more realistic training data and the neural optimizer becomes better and better at minimizing physics-based losses. Furthermore, since this way the training pool constantly generates new training data on the fly, its memory footprint is small enough such that the entire training pool can be kept in GPU memory. After training

for about 6 hours on a Nvidia Geforce RTX 4090, we obtained a single model that was able to solve various PDEs at high precision and produced all of the results shown in the following section.

## 5 RESULTS

In this section, we show that the very same model (same architecture and same weights) can deal with a wide variety of PDEs - linear as well as non-linear, stationary as well as time-dependent.

### 5.1 LINEAR SECOND ORDER PDES

Linear second order PDEs can be grouped into 3 categories: Elliptic PDEs (e.g., Poisson or Laplace equation), Parabolic PDEs (e.g., advection-diffusion equation) and Hyperbolic PDEs (e.g., wave equation). Here, we show that our model excels in all 3 categories:

#### 5.1.1 POISSON AND LAPLACE EQUATION

The Poisson equation is a stationary, elliptic PDE and is important to find e.g. minimal surfaces, steady state equilibria of the Diffusion-Equation (see Section 5.1.2) or to calculate energy potentials of electrostatic or gravitational fields (see Figure 12 in Appendix):

$$F = \Delta u - f = 0 \tag{5}$$

The Laplace Equation is the homogeneous ($f = 0$) special case of the Poisson equation.

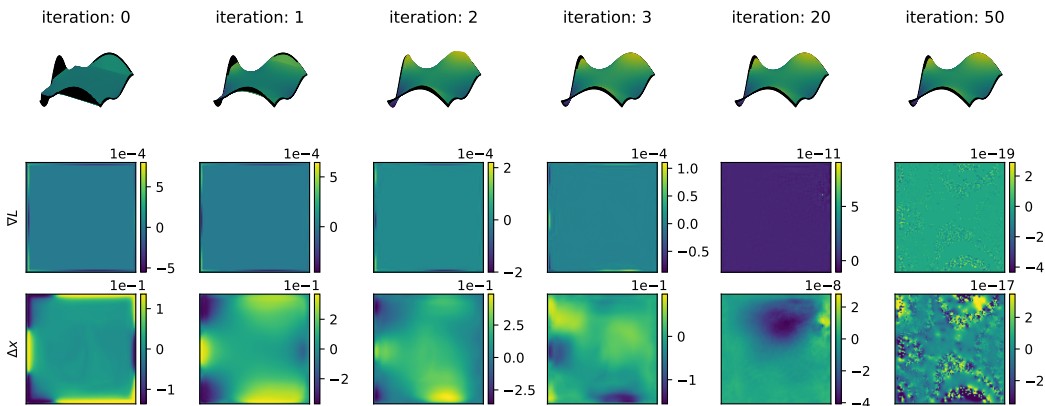

Figure 4: Iterative refinements by Metamizer to solve the Laplace equation. Top row: Intermediate solutions of $u$. Middle row: corresponding gradients of the physics-based loss. Bottom row: update steps computed by Metamizer. An animation of this process is presented in the supplemental video.

Figure 4 depicts intermediate solutions $u_i$ of the Laplace equation, the gradients of the physics-based loss $\nabla_u L_i$, and update steps $\Delta u_i$ computed by Metamizer at iterations 0, 1, 2, 3, 20 and 50. At the beginning, $u$ is 0-initialized. Thus, high gradients of $L$ only appear at the domain boundaries where the boundary conditions result in discontinuities of $u$. Nevertheless, Metamizer is able to anticipate better update steps that also affect the surroundings of the boundaries to achieve faster convergence. After only 3 update steps, the shape of $u$ is visually close to the final solution. After 20 iterations, the gradients of $L$ are already in the range of $10^{-7}$ and after 50 steps, Metamizer hits the limits of machine precision (see noise in $\nabla_u L_{50}$ in Figure 4) rendering further improvements impossible. Figure 5 a shows how Metamizer automatically adjusts the scaling $s_i$: At the very beginning, $s_0$ starts at a relatively small value (0.05) to avoid initial divergence. Then, for the next 5 iterations, Metamizer increases $s_i$ to lower $L$ more efficiently. After that, $s_i$ is steadily decreased for finer and finer adjustments. Remarkably, this learned behavior corresponds very well with our "natural" gradient descent intuition provided in Section 3. After around 50 iterations, when machine level precision is reached, no further progress is possible and $s_i$ remains at a constant very low level.

We compared our approach to several gradient descent based methods of the pytorch optimizer package (SGD, Adam, AdamW, RMSprop, Adagrad, Adadelta) and iterative sparse linear system solvers from CuPy (conjugate gradients, minres, gmres, lsmr). Figure 6 shows that traditional gradient based approaches are not competitive with current sparse linear system solvers regarding runtime or accuracy. Furthermore, these gradient based optimizers require tuning of hyperparameters such as learning rate or momentum (more information is provided in Appendix E). While Metamizer does not yet reach the performance of specialized and highly optimized algebraic multigrid solvers such as AMGX (see Appendix C), it reaches competitive performance to sparse linear system solvers like CG, MINRES or GMRES and reaches a MSR of $6.7 \times 10^{-33}$ after only $0.08s$ while maintaining the generality of gradient based approaches. Thus, it can be directly applied for example to nonlinear and nonsymmetric PDEs as well (see Section 5.2). On top of that, Metamizer shows good scaling behavior on larger grid sizes (see Appendix B).

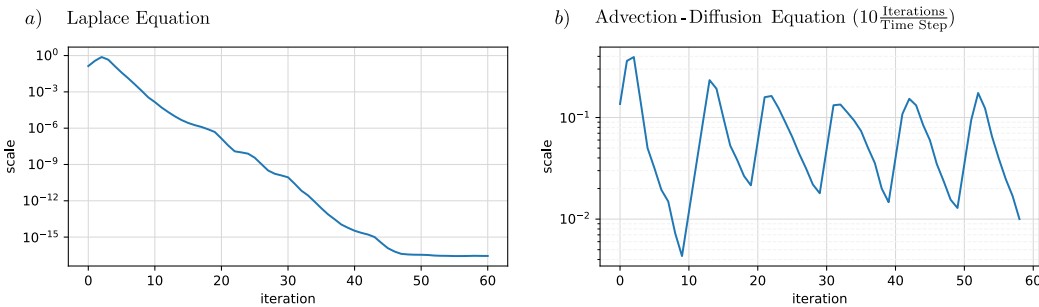

Figure 5: The scale parameter $s_i$ (see Figure 3 a) is automatically adjusted by Metamizer to make appropriate update steps. a) stationary Laplace Equation b) time-dependent advection-diffusion equation with 10 iterations per time step.

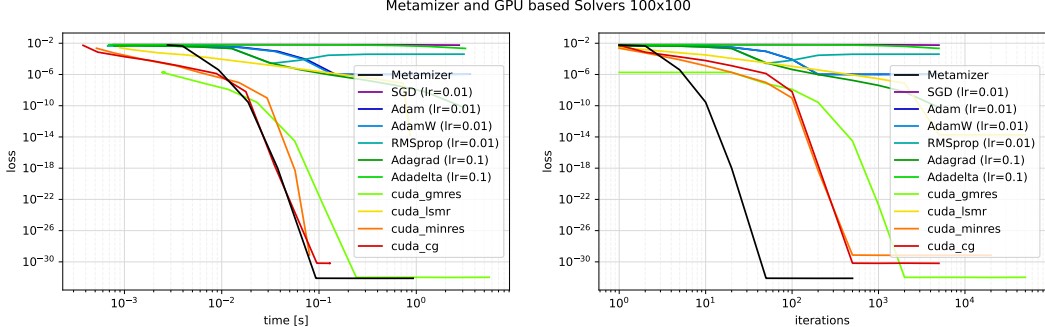

Figure 6: Comparison of mean squared residual errors for the Laplace equation on an $100 \times 100$ grid as a function of time (left) and number of iterations (right) between Metamizer and various iterative sparse linear system solvers of the CuPy package (Okuta et al., 2017) and gradient based optimizers from pytorch (various learning rates are investigated in Appendix E). The goal is to lower the loss in as little time as possible (bottom left corner of the left plot).

### 5.1.2 ADVECTION-DIFFUSION EQUATION

The Advection-Diffusion equation is an important parabolic PDE that describes for example heat transfer or the dispersion of pollutants. For an unknown function $u$, it can be written as:

$$F = \partial_t u - D\Delta u + \nabla \cdot (\vec{v}\,u) - R = 0 \qquad (6)$$

Here, $D$ denotes the diffusivity, $\vec{v}$ denotes the velocity field for the advection term and $R$ corresponds to the source / sink term. In contrast to the stationary Poisson equation, the advection-diffusion equation is time-dependent. Thus, we use the implicit backward Euler scheme for a stable unfolding of

the simulation in time. For every simulation time step, we perform a certain number of optimization steps to reduce $L$. By choosing the right number of iterations per time step we can make a trade-off between speed and accuracy. Figure 5 b) shows how Metamizer automatically adjusts the scaling $s_i$ when simulating the advection-diffusion equation at 10 iterations per time step. At the beginning of each time step, the neural optimizer increases its stepsize to quickly adjust $u$. Then, $s_i$ is decreased again for fine adjustments.

Figure 1 shows results for a constant localized source field $R$ and a rotating velocity field $\vec{v}$ (for example a laser that heats a spinning disk). More examples with different diffusivity parameters are shown in Appendix A. Table 1 shows how accuracy increases with the number of iterations per time step.

### 5.1.3 Wave equation

The wave equation is an important hyperbolic PDE that describes wave like phenomena such as water waves, pressure waves, electro-magnetic waves and many more. It can be written as:

$$F = \partial_t^2 u - c^2 \partial_x^2 u = 0 \tag{7}$$

Here, $c$ corresponds to the wave propagation speed. Figure 7 shows results for different values of $c$ that were obtained by Metamizer using an implicit Crank-Nicolson scheme. Note, that the wave equation was not included during training. Table 1 shows accuracy levels for 1, 5, 20 and 100 iterations per timestep. Using only 1 iteration results in divergent simulations, 20 iterations yield fairly accurate results and at 100 iterations, the simulation runs close to machine precision.

a) $c = 1$       b) $c = 2$       c) $c = 10$

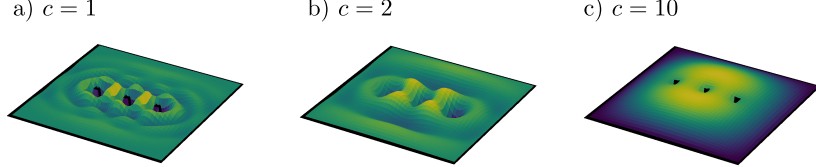

Figure 7: Metamizer is capable of simulating waves at different propagation speeds $c$ without having seen the wave equation during training.

### 5.2 Non-Linear PDEs

Non-linear PDEs are particularly hard to solve as the underlying systems of equations become non-linear as well. In this section, we demonstrate Metamizers capabilities to deal with non-linear PDEs.

### 5.2.1 Incompressible Navier-Stokes equation

The incompressible Navier-Stokes equation describes the dynamics of an incompressible fluid by means of a velocity field $\vec{v}$ and a pressure field $p$:

$$F = \rho \left( \partial_t \vec{v} + (\vec{v} \cdot \nabla) \vec{v} \right) + \nabla p - \mu \Delta \vec{v} - \vec{f}_{\text{ext}} = 0 \tag{8}$$

Here, $\rho$ denotes the fluids density, $\mu$ its viscosity and $\vec{f}_{\text{ext}}$ external forces. The incompressibility equation $\nabla \cdot \vec{v} = 0$ can be automatically fulfilled by utilizing a vector potential $a$ and setting the velocity field $\vec{v} = \nabla \times a$. However, using a vector potential prohibits us from directly manipulating the velocity field. Thus, we need an additional boundary loss term (Equation 4) to deal with the Dirichlet boundary conditions for the velocity field at the domain boundaries. To compute the partial derivatives of $a, \vec{v}$ and $p$ efficiently with centered finite differences, we rely on a staggered marker and cell (MAC) grid (Harlow et al., 1965; Holl et al., 2020; Wandel et al., 2021a).

Figure 8 shows results for various Reynolds numbers. The Reynolds number $Re$ is a unit less quantity defined as:

$$Re = \frac{\rho ||\vec{v}|| D}{\mu} \tag{9}$$

| PDE $\backslash \frac{\text{Iterations}}{\text{Time Step}}$ | 1 | 5 | 20 | 100 | $\frac{\text{Iterations}}{\text{Second}}$ |
|---|---|---|---|---|---|
| Advection-Diffusion | $7.9 \times 10^{-5}$ | $2.0 \times 10^{-7}$ | $5.6 \times 10^{-11}$ | $2.2 \times 10^{-33}$ | 420 |
| Wave Equation | $4.4 \times 10^{24}$ | $4.6 \times 10^{-4}$ | $3.2 \times 10^{-8}$ | $3.5 \times 10^{-31}$ | 420 |
| Navier-Stokes $L_p$ | $7.4 \times 10^{-6}$ | $2.0 \times 10^{-5}$ | $2.4 \times 10^{-8}$ | $2.4 \times 10^{-16}$ | 230 |
| $L_d$ | $1.7 \times 10^{-7}$ | $3.7 \times 10^{-7}$ | $9.5 \times 10^{-9}$ | $1.9 \times 10^{-17}$ | |
| 2D coupled Burgers | $4.1 \times 10^{-3}$ | $8.5 \times 10^{-6}$ | $2.5 \times 10^{-10}$ | $5.0 \times 10^{-32}$ | 330 |

Table 1: Mean Squared Residuals for different time-dependent PDEs at different iterations per time step

where $D$ corresponds to the diameter of the obstacle and $||\vec{v}||$ to the flow velocity. The Reynolds number has a big effect on the qualitative behavior of the flow field. For example, in case of very small Reynolds numbers, the flow-field becomes stationary and the Navier-Stokes equation can be approximated by the Stokes equation. In case of very high Reynolds numbers, the flow-field becomes turbulent and can be approximated by the Euler equation for inviscid fluids.

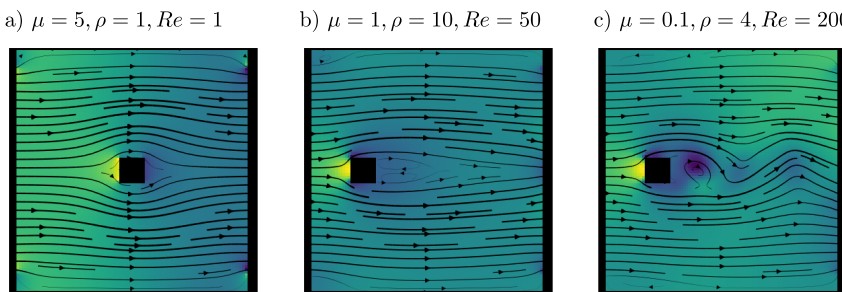

a) $\mu = 5, \rho = 1, Re = 1$      b) $\mu = 1, \rho = 10, Re = 50$      c) $\mu = 0.1, \rho = 4, Re = 200$

Figure 8: Fluids of various viscosities $\mu$ and densities $\rho$ at a wide range of Reynolds numbers can be simulated by Metamizer. The obstacle diameter is $D = 10$ and the fluid velocity is $||\vec{v}|| = 0.5$

Table 1 shows results for $L_p$ (mean squared residuals of Equation 8) and $L_d$ (mean squared residuals of $\nabla \cdot \vec{v}$ that capture if the Dirichlet boundary conditions are not properly met). $L_p$ and $L_d$ are lower for only 1 iteration compared to 5 iterations per time step, because 1 iteration per time step results in an incorrect steady-state solution for the wake dynamics. Compared to Wandel et al. (2021a), Metamizer reaches a similar performance at 5 iterations per time step. At 20 or even 100 iterations, our approach is orders of magnitudes more accurate while still being reasonably fast.

### 5.2.2 BURGERS EQUATION

The 2D coupled Burgers equation is an important non-linear PDE to study the formation of shock patterns.

$$F = \partial_t \vec{v} + (\vec{v} \cdot \nabla) \vec{v} - \mu \Delta \vec{v} = 0 \tag{10}$$

Here, $\mu$ is a viscosity parameter. Figure 1 shows an exemplary simulation result by Metamizer that exhibits clear shock discontinuities at $\mu = 0.3$. Additional qualitative results are shown in Figure 10 of Appendix A. Note that the Burgers equation was not considered during training. Mean squared residuals for different iterations per time step are shown in Table 1.

### 5.2.3 CLOTH DYNAMICS

To describe cloth dynamics, we closely follow the approach of Santesteban et al. (2022) and Stotko et al. (2024). Instead of using a classical PDE formulation, their approach relies on a spring-mass system consisting of a regular grid of vertex positions $\vec{x}$, velocities $\vec{v}$ and accelerations $\vec{a}$ that get integrated by a backward Euler scheme:

$$\vec{v}^t = \vec{v}^{t-1} + \Delta t \, \vec{a}^t \text{ and } \vec{x}^t = \vec{x}^{t-1} + \Delta t \, \vec{v}^t \tag{11}$$

To obtain the accelerations $\vec{a}^t$, the following physics-based loss $L$ must be minimized by Metamizer:

$$L = E_{\text{int}}(\vec{x}^t) - \Delta t^2 \langle \vec{F}_{\text{ext}}, \vec{a}^t \rangle + \frac{1}{2}(\Delta t)^2 \langle \vec{a}^t, M\vec{a}^t \rangle \tag{12}$$

Here, $\vec{F}_{\text{ext}}$ corresponds to external forces (such as gravity or wind), $M$ is the mass matrix and $E_{\text{int}}$ are the internal energies of the cloth specified as follows:

$$E_{\text{int}} = c_{\text{stiff}} \frac{1}{2} \sum_{\vec{e} \in \text{edges}} (||\vec{e}|| - 1)^2 + c_{\text{shear}} \frac{1}{2} \sum_{\psi \in \text{right angles}} (\psi - 90°)^2 + c_{\text{bend}} \frac{1}{2} \sum_{\theta \in \text{straight angles}} (\theta - 180°)^2$$

where $c_{\text{stiff}}, c_{\text{shear}}, c_{\text{bend}}$ are parameters that describe stiffness, shearing and bending properties of the cloth.

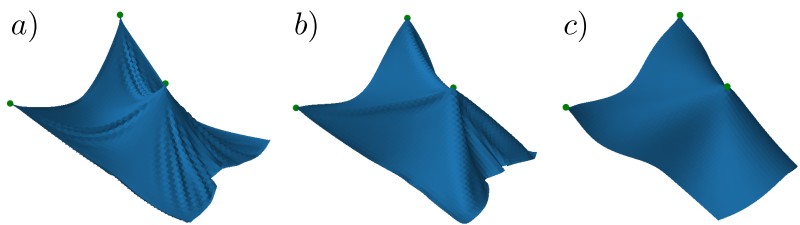

Figure 9: Cloth with various material parameters $(c_{\text{stiff}}/c_{\text{shear}}/c_{\text{bend}})$ can be simulated.
a) $(1000/10/0.01)$ b) $(1000/1000/10)$ c) $(1000/10/1000)$

Figure 9 shows results by Metamizer for various cloth parameters. The simulations were performed at 20 iterations per time step and around 185 iterations per second. Note that $L$ can take negative values and cannot be interpreted as easily as the mean squared residuals of the previous PDEs. Thus we ommited $L$ in Table 1 for our cloth simulations. Instead, we provide visualizations for the reduction of the gradient norm $||\partial_{\vec{a}} L||_2$ for different iterations per timestep in Appendix G.

## 6 CONCLUSION

In this work, we introduced Metamizer, a novel neural optimizer for fast and accurate physics simulations. To this end, we propose a scale-invariant architecture that suggests improved gradient descent update steps to speed up convergence on a physics-based loss at arbitrary scales. Metamizer was trained without any training data but directly on the mean square residuals of PDEs such as the Laplace, advection-diffusion or Navier-Stokes equations and cloth simulations in a meta-learning manner. Remarkably, our method also generalizes to PDEs that were not covered during training such as the Poisson, wave and Burgers equation. By choosing a proper number of iterations per timestep, a trade-off between speed and accuracy can be made. Our results demonstrate that Metamizer produces fast as well as highly accurate results across a wide range of linear and nonlinear physical systems.

In the future, further physical constrains such as Neumann or Robin boundary conditions and self-collisions for cloth-dynamics could be included. At the moment, Metamizer is trained for a single grid size. Thus, increased flexibility could be achieved by training Metamizer on multiple grid sizes simultaneously or by extending the network architecture to Mesh- or Graph-Neural Networks. For simulations on Mesh structures, a physics-informed loss based on Galerkin neural networks (Gao et al., 2022) could be practical. Furthermore, multiple PDEs such as the Navier-Stokes and advection-diffusion equation could be coupled.

We believe that our approach may have applications in future numerical solvers and speed up accurate simulations in computer games or computer-aided engineering. Since gradient descent approaches are ubiquitous, our approach may also serve as an inspiration for other gradient-based applications outside of physics, such as style transfer, scene reconstruction, or optimal control.

ACKNOWLEDGMENTS

This work was supported by the Federal Ministry of Education and Research of Germany and the state of North Rhine-Westphalia as part of the Lamarr Institute for Machine Learning and Artificial Intelligence and by the Federal Ministry of Education and Research under Grant No. PB22-063A INVIRTUO and Grant No. 01IS22094A WEST-AI. Furthermore, it is affiliated to VRVis GmbH in the scope of COMET - Competence Centers for Excellent Technologies (879730, 911654) which is managed by FFG.

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

## A   ADDITIONAL QUALITATIVE RESULTS

In this Section we provide additional qualitative results for Burgers equation with $\mu = 0.3/1$ (Figure 10), the advection-diffusion equation with $D = 0.1/0.5/2/10$ (Figure 11) and the Poisson equation (Figure 12) on a $100 \times 100$ grid. Furthermore, we trained Metamizer on a larger $400 \times 400$ grid to enable fluid simulations at higher Reynolds numbers. Figure 13 demonstrates that Metamizer is able to simulate complex turbulent fluid dynamics at $Re = 2000$.

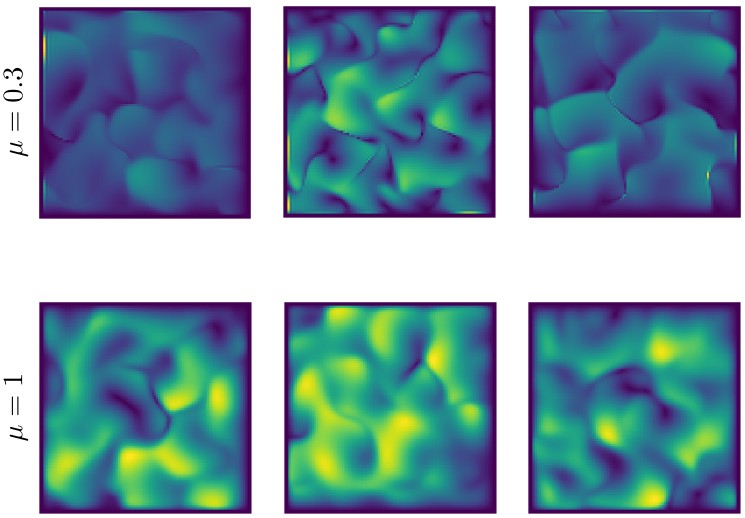

Figure 10: Burgers equation with $\mu = 0.3/1$

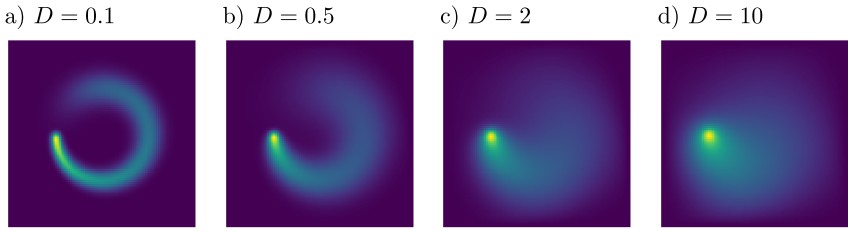

Figure 11: Advection-Diffusion equation with $D = 0.1/0.5/2/10$. (Domain Boundaries have Dirichlet BC = 0.)

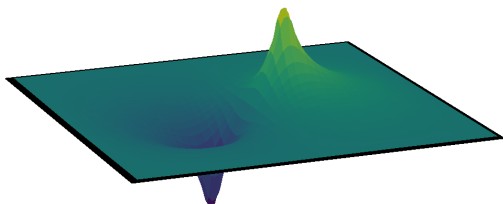

Figure 12: Solution of the Poisson equation for the electrostatic potential of two oppositely charged particles within a box with 0-boundary conditions. Note, that Metamizer was only trained on the Laplace equation ($f = 0$) while, here, $f \neq 0$ at the particle positions.

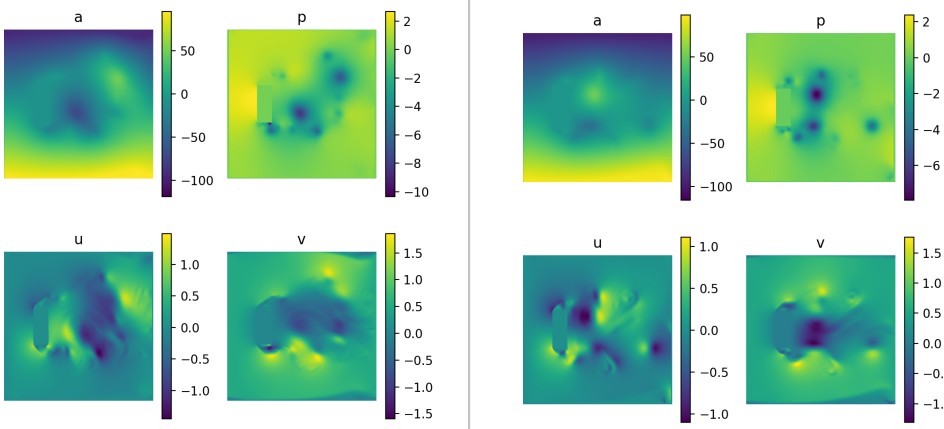

Figure 13: Simulation of turbulent fluid dynamics at $Re = 2000$ ($\mu = 0.1, \rho = 4, D = 100, ||\vec{v}|| = 0.5$) performed by Metamizer on a $400 \times 400$ grid.

## B  DOMAIN-SIZE DEPENDENT PERFORMANCE

In order to investigate the domain size dependent performance scaling of Metamizer, we trained another network on a $400 \times 400$ grid to solve the Laplace equation. In the following, we compare Metamizers performance scaling to GPU and CPU based solvers.

### B.1  COMPARISON TO GPU BASED SOLVERS

We run Metamizer against various GPU based solvers of the CuPy package on the same Nvidia Geforce RTX 4090. As shown in Figure 6, Metamizer shows a similar convergence speed for the Laplace equation on a $100 \times 100$ grid. On a larger $400 \times 400$ grid (see Figure 14), Metamizer demonstrates improved scaling behavior compared to sparse linear system solvers resulting in about $2\times$ faster convergence than conjugate gradients or MINRES.

This is remarkable because Metamizer is a more general optimizer that relies only on local gradients and can be used to solve for example nonlinear PDEs or cloth simulations as well. Conjugate gradient methods on the other hand require full knowledge of the underlying (symmetric) matrix $A$ to orthogonalize update steps and compute appropriate step sizes $\alpha$.

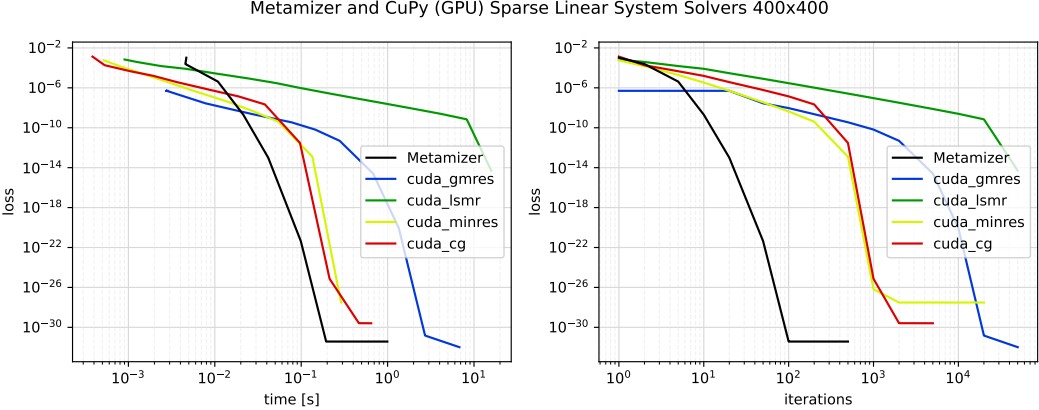

Figure 14: Performance comparison of Metamizer with various GPU based solvers of the CuPy package on a $400 \times 400$ grid. Metamizer is about $2\times$ faster than CG on the same GPU.

## B.2 Comparison to CPU based solvers

We compared Metamizer to CPU based solvers as well. On a $100 \times 100$ grid, CPU based solvers are actually slightly faster than GPU based solvers and Metamizer (see Figure 15). This is because CPUs exhibit a smaller computational overhead to GPUs and run at a higher clockrate. However, on a larger $400 \times 400$ grid, GPU parallelization pays off significantly resulting in $10\times$ faster convergence speed of Metamizer compared to CPU based solvers (see Figure 16).

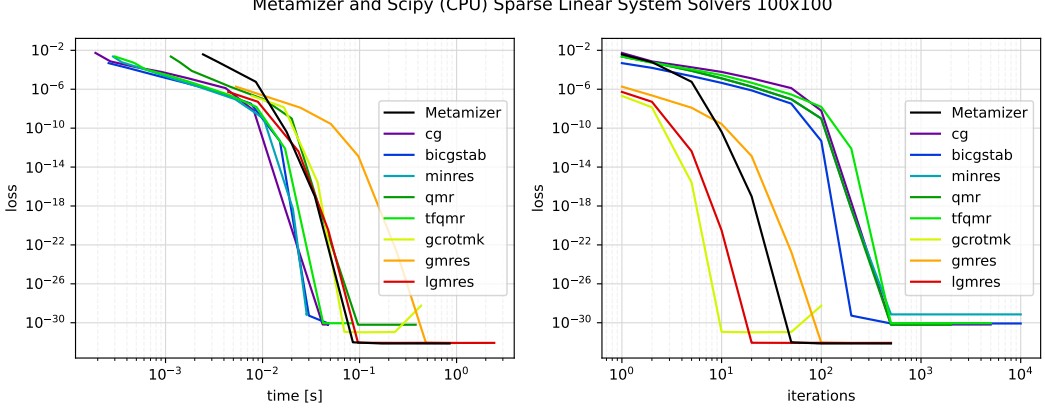

Figure 15: Performance comparison of Metamizer with various CPU based solvers of the SciPy package on a $100 \times 100$ grid.

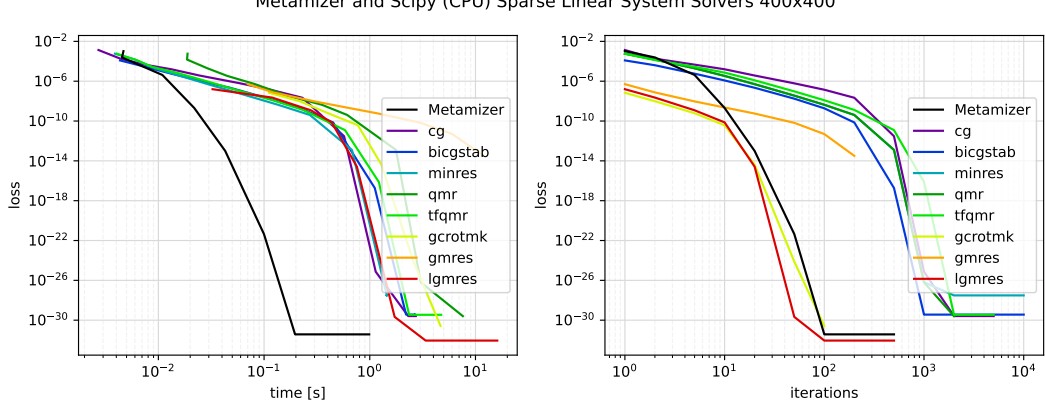

Figure 16: Performance comparison of Metamizer with various CPU based solvers of the SciPy package on a $400 \times 400$ grid. Metamizer is about $10\times$ faster than CG on a CPU.

## C Preconditioners

### C.1 Incomplete LU (ILU) and Algebraic Multi Grid (AMG) Preconditioners

We compared Metamizer with preconditioned conjugated gradient methods using incomplete LU factorization and algebraic multigrids. To this end, we relied on pyAMG (Bell et al., 2023) for a CPU based implementation and pyAMGX based on Nvidia AMGX. AMG preconditioning resulted in a significant speed-up over non-preconditioned solvers on the CPU as well as the GPU (see e.g. Figure 17 and Figure 18). While Metamizer is still faster than CPU based AMG methods, the highly optimized AMGX package by Nvidia is up to 10 times faster than Metamizer on the GPU. For domains of this size, this performance difference is in accordance with previous results for neural preconditioners (see Figure 4 in Lan et al. (2023)). However, Metamizer is a more general approach

compared to AMG approaches and neural preconditioners since it only requires local gradients and can be applied for example to non-symmetric Matrices, nonlinear PDEs or cloth simulations as well.

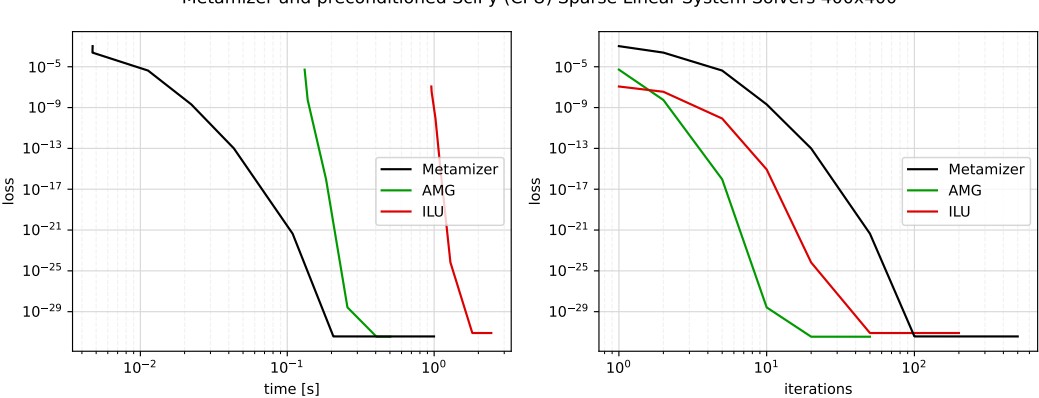

Figure 17: Performance comparison of Metamizer with conjugated gradients preconditioned on the incomplete LU factorization and the algebraic multigrid method (pyAMG (Bell et al., 2023)) on a 400x400 grid.

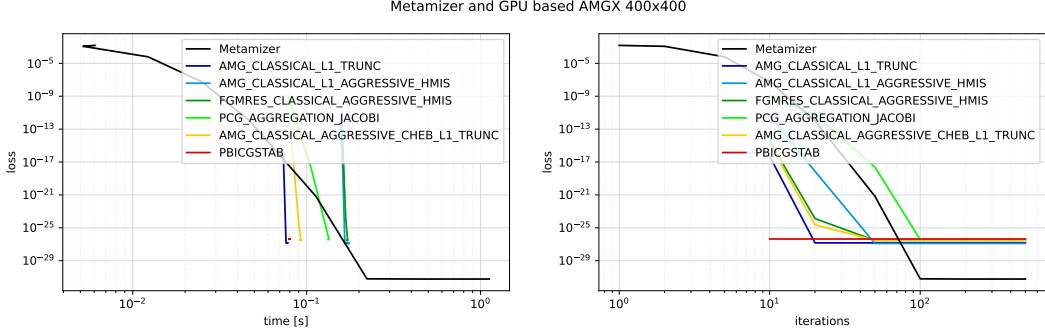

Figure 18: Performance comparison of Metamizer with algebraic multigrid methods based on Nvidia AMGX (pyamgx) on a 400x400 grid. Note: the plotted timings of AMGX include initialization overheads. The pure setup and solve timings until convergence were $0.019s$ for AMG_CLASSICAL_L1_TRUNC, $0.1s$ for AMG_CLASSICAL_L1_AGGRESSIVE_HMIS, $0.1s$ for FGMRES_CLASSICAL_AGGRESSIVE_HMIS, $0.069s$ for PCG_AGGREGATION_JACOBI, $0.028s$ for AMG_CLASSICAL_AGGRESSIVE_CHEB_L1_TRUNC and $0.035s$ for PBICGSTAB. Metamizer reaches a loss of $6.22 \times 10^{-32}$ after $0.22s$.

## C.2   JACOBI PRECONDITIONER

We tested the diagonal Jacobi preconditioner for CG, GMRES and MINRES. Since the diagonal of the Laplace operator on a regular grid corresponds closely to the identity matrix, this preconditioner did not help to significantly reduce the number of iterations but resulted in a slight computational overhead and thus a slowdown of convergence (see Figure 19).

## D   COMPARISON TO NEWTON-CG, NONLINEAR CG AND L-BFGS-B

We added further comparisons for optimizers that, like Metamizer, rely only on local gradients of the loss. As can be seen in Figure 20, optimizers like Newton-CG, Nonlinear CG or L-BFGS exhibit much slower convergence.

Metamizer and CuPy (GPU) Sparse Linear System Solvers (Jacobi Preconditioner) 400x400

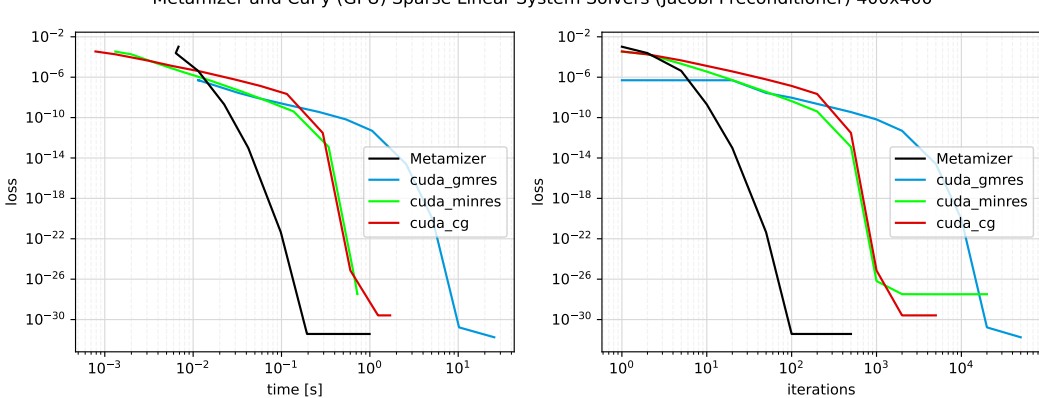

Figure 19: Performance comparison of Metamizer with various GPU based solvers of the CuPy package that make use of a Jacobi Preconditioner on a 400x400 grid. In comparison to non-preconditioned solvers (Figure 14) this resulted in a slight slowdown of convergence.

Metamizer and Scipy (CPU) Optimizers 100x100

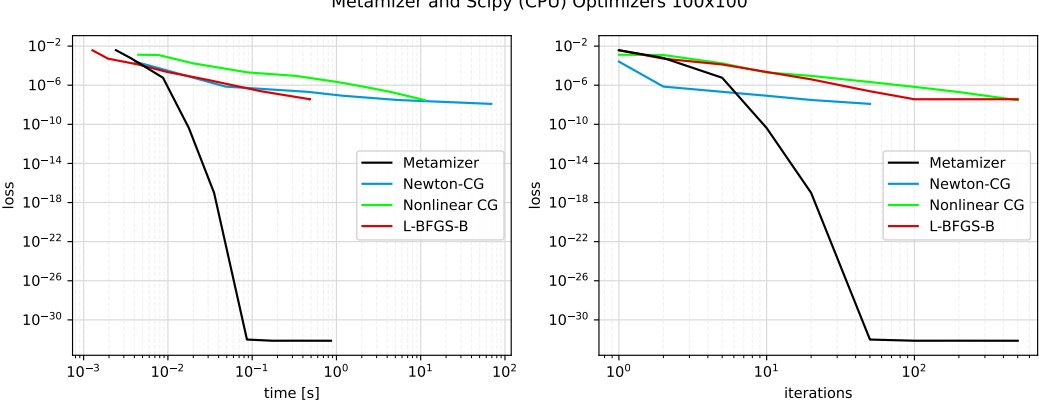

Figure 20: Performance comparison of Metamizer with various optimizers that, like Metamizer, rely only on local gradients of the loss. Here, we make use of the SciPy package on a $100 \times 100$ grid on a CPU.

## E    COMPARISON OF DIFFERENT LEARNING RATES

We compared various different learning rates for all gradient descent solvers and only reported results for learning rates that resulted in good trade-offs between convergence speed and accuracy in Figure 6. Figure 21 shows a performance comparison of Metamizer with Adagrad, Adam and AdamW at different learning rates.

## F    COMPARISON TO GROUND TRUTH

We compared the mean squared errors of Metamizer and various GPU based linear system solvers to analytical ground truth values of the Laplace equation. Figure 22 shows fast convergence to highly accurate results as already indicated by the mean residual errors in Figure 14.

## G    RESIDUAL LOSS GRADIENTS FOR CLOTH SIMULATIONS

Figure 23 shows how Metamizer reduces the gradient norm $||\partial_{\vec{a}} L||_2$ of the physics-based Loss defined in Equation 12 with respect to the accelerations of the cloth for 10, 20 and 50 iterations per timestep. With 10 iterations, Metamizer reaches a gradient norm of around $3 \times 10^{-2}$, with 20

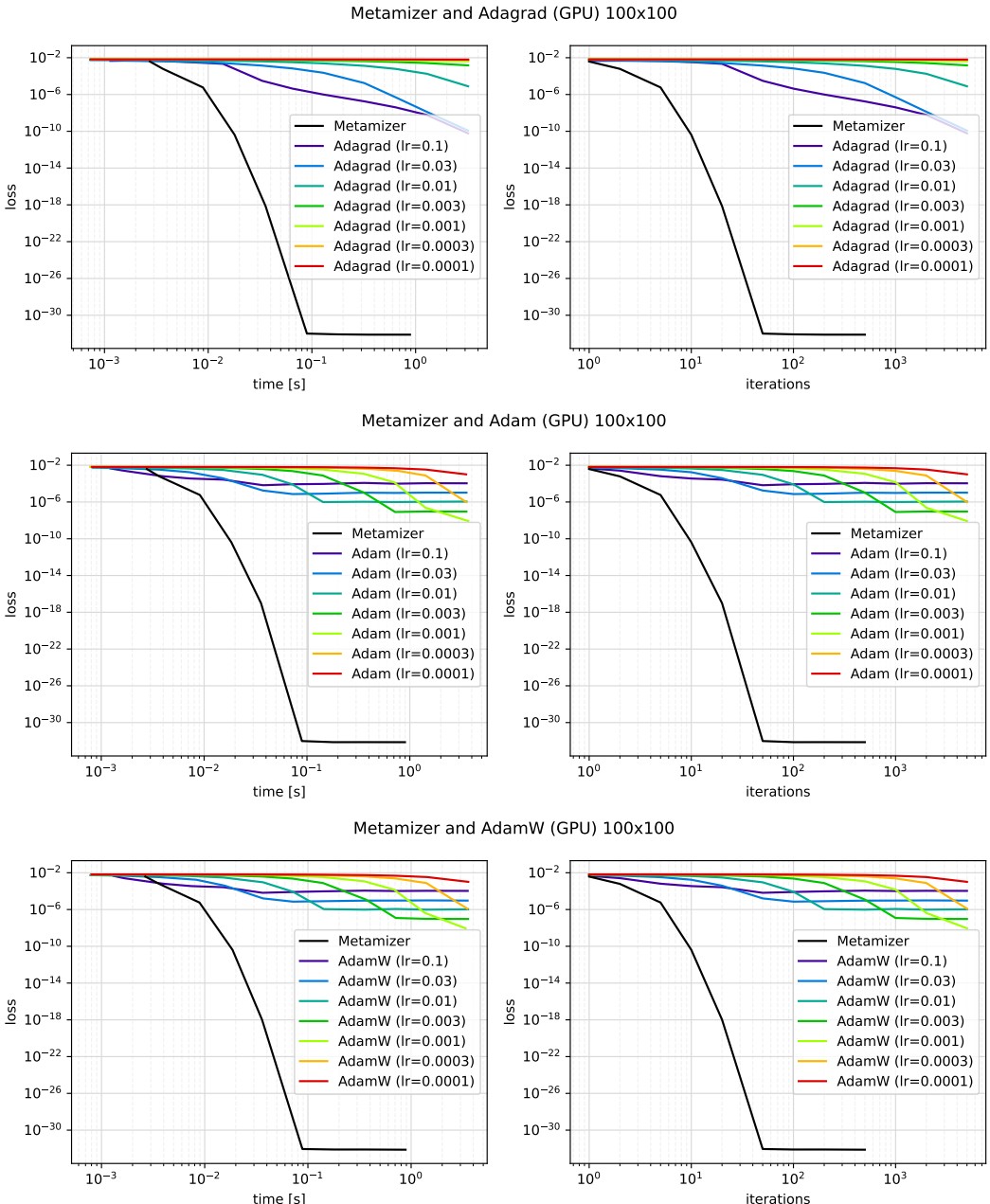

Figure 21: Performance comparison of Metamizer with various GPU based Gradient Descent Methods at different learning rates on a $100 \times 100$ grid.

iterations, it reaches a gradient norm of around $1 \times 10^{-2}$ and with 50 iterations, it reaches a gradient norm of around $4 \times 10^{-3}$.

## H  SCALING PARAMETERS

In Figure 24, we show additional results how Metamizer scales $s_i$ for $3, 5, 10, 20, 50, 100$ iterations per timestep on the Advection-Diffusion equation. For 50 and 100 iterations, a significant amount of iterations is "wasted" to scale up $s_i$ for the next timestep. This could be improved in the future by

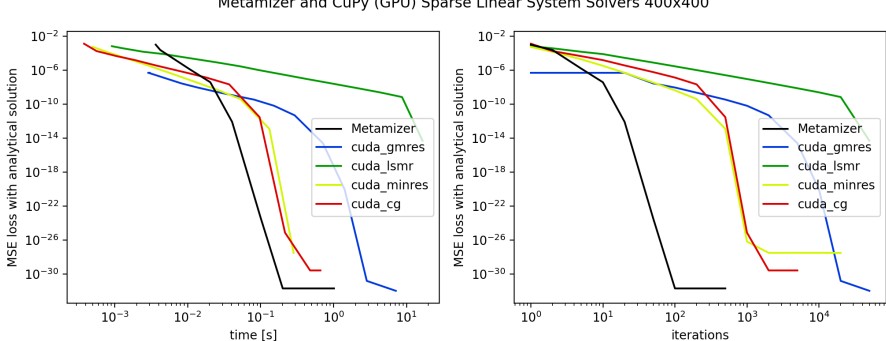

Figure 22: Mean squared errors of Metamizer and various GPU based solvers of the CuPy package on a $400 \times 400$ grid compared to analytical ground truth values of the Laplace equation.

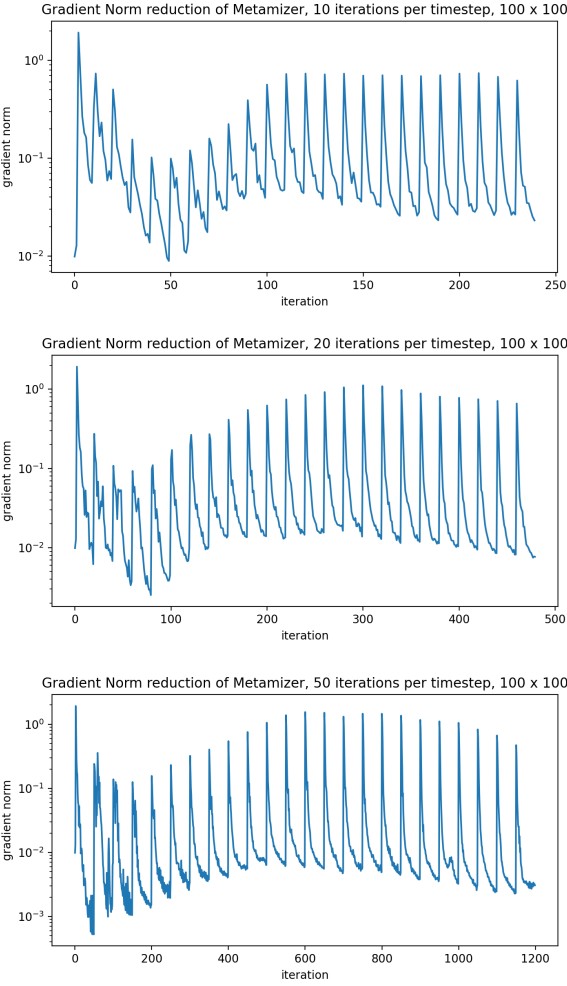

Figure 23: Metamizer iteratively decreases the gradient norm of a physics-based loss for nonlinear cloth simulations on a $100 \times 100$ grid. The more iterations per timestep are taken the more accurate is the simulation.

automatically scaling $s_i$ to a certain percentage (e.g. 20 %) of the maximum scale of the previous time step when starting with a new time step.

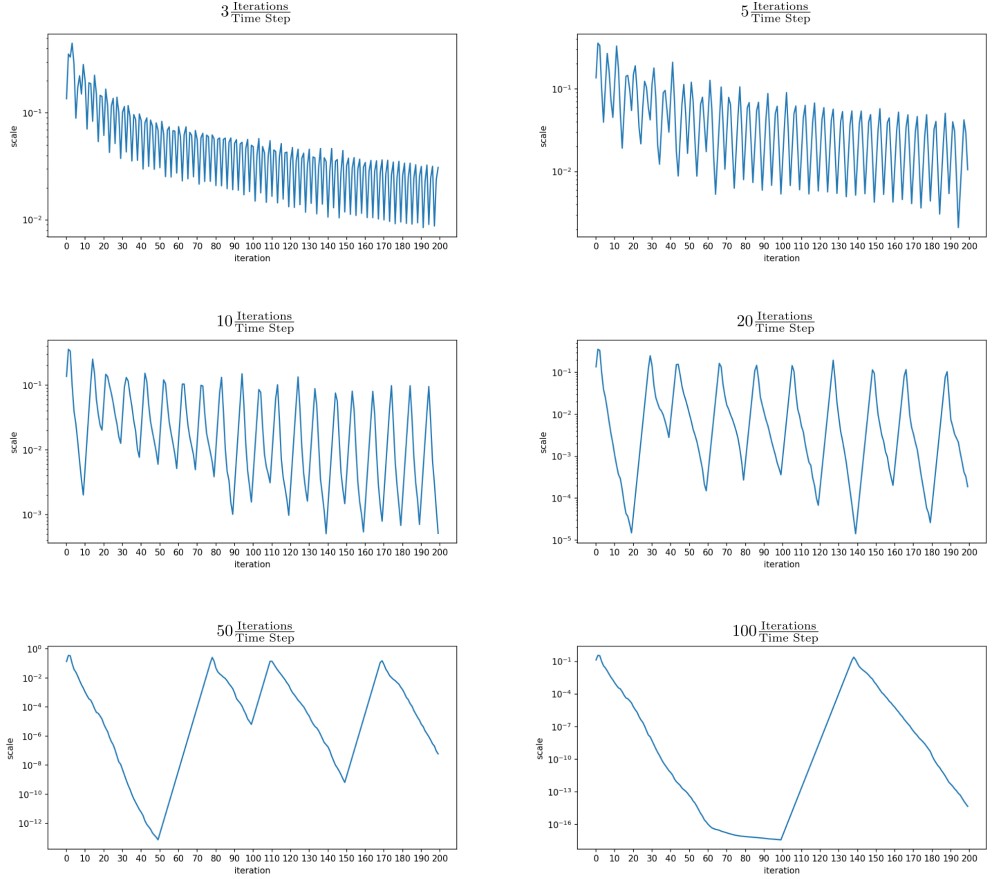

Figure 24: Scaling parameters for Advection-Diffusion for different numbers of iterations per time step

# I  LAPLACE OPERATOR DETAILS

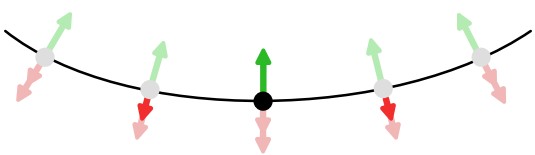

Figure 25: Implementation detail for Laplace operator - example in 1D

During our experiments, we found that naively applying the finite-difference Laplace operator results in very small gradients - even if the residuals are still quite large. Intuition on this problem is provided in Figure 25: If we want to lower the Laplacian for the center vertex (marked in black), we could increase the value of that vertex (marked in green) but we could also decrease the values of its neighboring vertices (marked in red). The same is true for all vertices, such that in the end the gradients pointing upwards get canceled out fairly exactly be the neighboring gradients pointing downwards. To avoid this issue, we detach (and thereby remove) all the neighbor-gradients (marked in red) so that we are only left with the green gradients and can make much better progress in solving the Laplace operator.

## J    TRAINING POOL DETAILS

The training pool strategy as described in Section 4 does not require any precomputed training data. Instead, Metamizer learns to solve the PDEs purely based on the physics-constrained loss and randomized boundary conditions that can be generated on the fly similar to Wandel et al. (2021a), Stotko et al. (2024) or Geneva & Zabaras (2020). For example, in case of fluid simulations we generate random boundaries by placing randomly moving boxes within the fluid domain, randomizing $\mu$ and $\rho$ or changing the flow velocity. In case of cloth simulations, we randomize cloth parameters such as $c_{\text{stiff}}$, $c_{\text{shear}}$ and $c_{\text{bend}}$ as well as the points at which the cloth is fixed. For the advection-diffusion equation, we randomized the diffusivity parameter $D$ as well as the velocity field $\vec{v}$ and the domain boundary. Full details about the different randomization strategies will be provided with our code. After training, Metamizer shows good performance across a wide range of PDEs and even generalizes to PDEs that were not covered during training such the wave, Poisson or Burgers equation.

## K    VIDEO

The supplementary video demonstrates how Metamizer solves the Laplace, diffusion-advection, wave, Navier-Stokes and Burgers equations as well as cloth simulations. The time-dependent PDEs are visualized at the speed of the simulation.

