# OpenReview forum: "Metamizer: A Versatile Neural Optimizer for Fast and Accurate Physics Simulations"
_ICLR.cc/2025/Conference — ICLR 2025 Poster_

### Official Review · Reviewer_8DMR · 2024-10-26

**Soundness:** 2
**Presentation:** 3
**Contribution:** 2
**Rating:** 5
**Confidence:** 4

**Summary:**

The paper studies solving partial differential equations (PDEs) numerically. The proposed framework builds upon a finite-difference spatial discretization of the unknown fields on a regular grid. Next, it minimizes a residual loss similar to that in the physics-informed neural network (PINN) via gradient-based optimization. The new idea comes in a neural network that predicts the update rule (scaling factor and direction) used in each step of the gradient-based optimization. The paper demonstrates this idea on several linear and nonlinear PDE problems.

**Strengths:**

- The idea of applying meta-optimizers to learning-based PDE solvers looks new to me. I think this is an interesting thought worth further discussion and exploration.
- The technical method is simple, straightforward, and easy to implement.
- The paper writing is good; it articulates its technical details very well.
- The paper considers a diverse set of PDE problems, including N-S equations and 3D cloth simulation (albeit with limited features).

**Weaknesses:**

**Introduction/Related Work**
- The paper claims numerical methods for solving PDEs “are computationally
expensive”/“are still relatively expensive to compute”/“require high computational resources.” This argument is generally valid for large-scale problems. However, for the small-sized problems demonstrated in this work (400 x 400), Chances are that numerical methods are not as expensive as the paper indicates and may be faster than Metamizer (See Poisson/Laplacian problem below). I suggest the paper evaluate its method on problems of much larger sizes, e.g., on >=1024^2 (2D) or >=64^3 (3D) grids.

**Foundations**
- Equations 3 and 4 are not new and should cite the original PINN paper. It is probably also worth mentioning that minimizing Equation 3 may get stuck in a local minimum (grad L = 0, but L is nonzero), which is not a solution to the original PDE. For several problems (linear second-order PDEs) demonstrated in the experiments, a variational loss from the PDE’s weak form is a standard and a naturally better candidate for the loss design. The incremental potential in Eqn. 12 is one such example that highlights this point.
- The issue in Fig. 2 and the main intuition for taking gradients from previous steps into account have been extensively studied in numerical optimization methods. The former is typically handled with (exact) line searches, and the latter is discussed in (nonlinear) CG, L-BFGS, etc. (“Numerical Optimization” by Nocedal & Wright.) It is still very interesting to contemplate the possibility of training a neural network to exploit gradients from previous steps, but the paper should discuss its relation to these classic numerical optimization methods and compare them in an experiment.

**Experiments**
- The Poisson/Laplacian example has the most complete analysis, but the main results in Fig. 6 look dubious for several reasons:
1) It looks like Metamizer has used an Nvidia RTX 4090 GPU (line 330 and Appendix B), so it should be compared with linear solvers implemented on the same GPU. However, I don’t recall scipy having a GPU implementation of them. Please correct me if I am wrong.
2) The state-of-the-art GPU implementation of a Krylov subspace solver (CG, GMRES, etc.) is likely to be better than both the scipy baselines and Metamizer reported in Fig. 6 and Appendix B. Back in 2005, deploying a standard CG algorithm on a GPU could solve wave/Poisson equations on a 1024^2 grid at a rate of 10-30 fps (Chapter 44 in “GPU Gems 2” edited by Pharr & Fernando and the referenced papers), already arguably faster than the speed of Metamizer reported in the much smaller 400^2 problem. I feel this paper didn’t identify the right baseline for comparison.

- The wave equation example does not compare Metamizer’s performance with a baseline. I suggest comparing it with the CG-GPU baseline mentioned above.

- Eqn. 12 in the cloth example is not new and should cite its original source, e.g., Stotko 2024. This equation can be traced back to at least “Optimization Integrator for Large Time Steps” by Gast et al. in 2015, if not earlier.

- “Note that L can take negative values and cannot be interpreted as easily as the mean squared residuals of the previous PDEs. Thus we omitted L in Table 1 for our cloth simulations.” In this case, grad L = 0 is the right indicator for successfully solving the PDE. There should be a table reporting grad L after certain iterations.

- On the algorithmic side, I feel some classic approaches for exploiting gradients from past iterations should be considered for comparison to establish the advantage of the network module in Fig. 3a. Two candidates are nonlinear CG and L-BFGS (with history size = 2). These methods do not require training and can be directly used to minimize Eqns. 3 and 4.

**Questions:**

- Does CG fail to converge in Fig. 6? Is the matrix not symmetric positive definite?

---

> ### Author Response · Authors · 2024-11-19
>
> Dear reviewer,
> Thanks for your questions and remarks!
>
> Regarding the mentioned weaknesses:
>
> **Introduction:** Of course, computational expenses must be always seen in context of other approaches. Since our baselines were the biggest point of critique, we added several further GPU baselines (including CG, GMRES, MINRES) that indicate that Metamizer remains the fastest method on a 400x400 grid (see Section 1 in Rebuttal.pdf in Supplementary). We hope that our work can make a good contribution to reduce the computational workload for a wide range of physics simulations which is still a very active field of research.
>
> **Foundations 1:** We cite several works on PINNs. Equations 3 and 4 are very basic ML knowledge that dates back at least to "Artificial Neural Networks for Solving Ordinary and Partial Differential Equations" by Lagaris et al (1997). Local minima in $L$ did not cause any problems in our experiments (as can be seen by the residuals $\approx 0$ in Table 1). While the weak form is often applied for example in finite element methods, here, we rely on finite differences and follow the most common approach of PINNs based on a residual loss.
>
> **Foundations 2:** Line search strategies would be expensive for nonlinear PDEs. We added further Nonlinear CG and L-BFGS-B baselines (see Section 3 in Rebuttal.pdf) that clearly show inferior convergence behavior compared to the previous baselines already provided. We’ll add a corresponding discussion.
>
> **Experiments 1:** Indeed, GPU baselines were missing in the current version. Thus, we now added further comparisons to GPU based sparse linear system solvers (including CG, GMRES, MINRES) using the CuPy package (see Section 1 in Rebuttal.pdf). This indeed significantly improved the baselines on the 400x400 grid, but Metamizer still remains the fastest method. Unfortunately, [Chapter 44 in GPU Gems 2](https://developer.nvidia.com/gpugems/gpugems2/part-vi-simulation-and-numerical-algorithms/chapter-44-gpu-framework-solving) does not provide an in depth analysis of the achieved accuracy or the number of CG iterations used. Thus, a direct comparison is not possible. We assume they only used a few iterations to achieve the desired visual effects.
>
> **Experiments 2:** We added several GPU baselines for the Laplace equation (see Section 1 in Rebuttal.pdf). For other linear PDEs (e.g. wave equation), we expect similar performance.
>
> **Experiments 3:** We acknowledge Santesteban et al. (2022) and Stotko et al. (2024) at the very beginning of the section (Line 481,482).
>
> **Experiments 4:** We’ll add a Table that reports gradient norms in the supplementary.
>
> **Experiments 5:** Thanks for the remark. We added nonlinear CG and L-BFGS baselines for a better comparison (see Section 3 in Rebuttal.pdf).
>
> Regarding the question on the convergence of CG:
> Indeed, CG didn't converge since the matrix wasn't symmetric due to our naive implementation of the Dirichlet boundary conditions. We already fixed that (see Section 2 in Rebuttal.pdf). While on small grids ($100\times 100$), this led to faster convergence, Metamizer is still far ahead on larger grids ($400\times 400$).

---

> > ### Comment · Reviewer_8DMR · 2024-11-25
> > **Thank you for the response**
> >
> > Thank you for the new experiments that compared your method with iterative solvers on GPUs. While I am still not fully convinced with the GPU CG performance reported in the new experiments, I will raise my score (3 -> 5) because I think exploring this direction of learning + optimization should be encouraged.
> >
> > It would be great if the paper could assess its performance on larger Poisson problems typically evaluated in previous works (e.g., 128^3 or 1024^2 as I mentioned in my review) and calibrate its CG baselines with known statistics/implementations, e.g.,
> >
> > - Fig. 6 in https://arxiv.org/pdf/2310.00177;
> > - https://docs.nvidia.com/cuda/incomplete-lu-cholesky/index.html#algorithm-1-conjugate-gradient-cg;
> > - It is also helpful to check out the two reference papers in the GPU Gems link I mentioned.

---

> > > ### Author Response · Authors · 2024-11-26
> > >
> > > Dear reviewer,
> > >
> > > Thanks for your encouraging reply and acknowledging the improvements of our submission!
> > >
> > > While Section 6 in the Rebuttal.pdf already gives a glimpse at larger turbulent fluid simulations by Metamizer on a 400x400 grid, unfortunately, the short-term deadline doesn’t allow us to set up, run and evaluate new experiments on an even larger 1024x1024 grid. Nevertheless, we hope that our work can serve as a proof of concept for future large scale simulations.
> > >
> > > Regarding the references to more sophisticated preconditioners for conjugate gradients: We added further comparisons with incomplete LU and algebraic multigrid preconditioners using pyAMG (see Section 7.1 in Rebuttal.pdf). In particular, AMG showed greatly improved convergence speed. Unfortunately, we were not able to run these preconditioners on a GPU based library in time and we suppose that an optimized GPU implementation of AMG or neural preconditioners as proposed by Kaneda et al. indeed could outperform Metamizer on the Poisson equation. So we'll be more careful with our framing of Metamizer and focus more on benefits regarding its versatility with respect to non-symmetric/non-linear PDEs in our revised manuscript.

---

> > > > ### Comment · Reviewer_8DMR · 2024-12-03
> > > > **Official comment by the reviewer**
> > > >
> > > > Thank you for the update. I will maintain my current score.

---

### Official Review · Reviewer_shqZ · 2024-10-28

**Soundness:** 3
**Presentation:** 3
**Contribution:** 3
**Rating:** 8
**Confidence:** 4

**Summary:**

This paper targets an optimizer for training deep neural networks for improving the solving of tasks related to PDE solving. The goal if scale invariance is identified as a central challenge for physics-related learning tasks. The paper then proposes a U-net that is employed to infer the scaling of an update in order to improve conditioning of the neural network update.

The importance of scale invariance has been studied before for physics problems, e.g. in Holl et al., Scale-invariant learning by physics inversion, NeurIPS 2022. I just double checked , and e..g fig 1 there and figure 2 here are actually have a lot in common. This previous paper in the end takes a different approach, so I think it's no direction competition, but the unclear parts of the current submission definitely lead to a few questions.

The paper evaluates this idea for several canonical PDEs (Poisson & Laplace, advection-diffusion), as well as more complicated ones (waves, Navier-Stokes). Interestingly, cloth dynamics are included as well. While the rang of experiments is very nice, the motivation in terms of theory behind the approach remains unclear. Nonetheless, I think it’s an interesting and novel direction. So I liked the paper and its direction, but several unclear topics remain, which I hope the authors can address during the rebuttal phase.

**Strengths:**

The central goal of addressing "scale invariance" is not new, but the proposed learning of scale invariance is new if I understood the approach correctly.

The breadth of results is also great to see.

**Weaknesses:**

One central part that was unclear to me was how the scale factors for learning are actually precomputed. Is this simply a scaling of the reference solutions from the solver? As this is the central component that is supposed to make the method work, I think it would be important to be clear where this comes from. Correspondingly, I was missing an evaluation in terms of how well the network actually learned to predict these scaling factors. The results seem to only show the loss in terms of the state.

The Holl'22 paper above also provides some theory on how updates from scale invariant solvers improve learning. Unfortunately the submission here does not take up on these topics, and focuses on experiments.

The appendix is also very brief on how the different learning tasks were actually set up.

**Questions:**

Closely related to the topics above, can the authors shed light on the following topics?

- How does the approach differ from the scale-invariant learning proposed in the NeurIPS'22 paper above?

- Does the theory there affect or relate to the current submission?

- How are the scaling factors s_i computed, and how accurately are they inferred?

- Have the authors considered Newton methods, that approximate the inverse Hessian for "scale invariance"? I think this would also be worth a discussion, at least.

- The paper focuses on solving optimization problems in terms of PDE states. How would this extend to a second neural network, i.e. backpropagating further than just the u?

---

> ### Author Response · Authors · 2024-11-19
>
> Dear reviewer,
> Thanks a lot for the overall positive review!
>
> Regarding:
>
> *The paper “ Scale-invariant learning by physics inversion” by Holl et al.:*
> This work is also motivated by scale-invariance of the loss function. However, it focuses on inverse problems in physics while we focus on physics simulations and solving PDEs. This results in very different approaches: While Holl et al. take a hybrid training approach to improve a gradient-descent based learning pipeline by embedding updates from a scale-invariant inverse problem solver, we achieve scale-invariance for physics simulations by using an internal scaling mechanism and normalizing the input gradients and step sizes (see Figure 3a). This allows our network to directly make scale-invariant update step predictions.
>
> *Scaling factor:*
> The scaling factor $s$ is an internal state of Metamizer that helps to better adjust the update steps (similar to a learning rate). Metamizer learns to increase and decrease $s$ automatically without any ground truth data. Thus, the performance / accuracy on $s$ can not be evaluated individually. However, the overall convergence behavior of Metamizer (see Figures 6 and 13) as well as the scaling investigations in Figures 5 and 14 indicate that Metamizer learns a reasonable scaling behavior.
>
> *Brief Appendix:*
> We’ll add further details in our appendix. Furthermore, we’ll release our code and a pretrained model so our results can be reproduced.
>
> *Newton Methods:*
> We added further comparisons to Newton-CG (see Section 3 in Rebuttal.pdf in the Supplementary)
>
> *Future neural network extensions:*
> Propagating gradients further than u could be an interesting direction of research in the future as it might enable applications for inverse problems.

---

> > ### Comment · Reviewer_shqZ · 2024-11-24
> > **Re: Official Comment**
> >
> > I’d like to thank the authors for the additional explanations and experiments. Good to know the method _does_ learn the scaling factors without being provided a reference. It’s also good to see the new GPU experiments. They do provide a fairer comparison with the other solvers
> >
> > I don’t fully agree with the comments about the scale-invariant inverse solves, though. Whether the PDE is “forward” or “backward” is a matter of definition to some-extent, and the goal remains the same: addressing the scaling issues. Scale-invariant solves from the NeurIPS paper above could actually provide some ground truth values for the scaling. This could build “trust” for the proposed method (if it turns out similar factors are learned).
> >
> > Overall, and despite the other more negative reviews, I still think this is a new and interesting direction for using NNs to solve PDEs. I also agree with some of the author’s comments above that PINNs are clearly far still away from “realtime” solves (and practical use cases). For me, that’s another good reason to encourage new ideas in the area. I think the combination of a new idea with the very good performance of the method is a good reason for an accept. To reflect this, I’ve raised my score by 1 step.

---

> > > ### Author Response · Authors · 2024-11-26
> > >
> > > Dear reviewer,
> > >
> > > Thanks a lot for your encouraging reply and acknowledging the improvements of our submission!
> > >
> > > You’re right that scale-invariance and in particular its mathematical formulation remains basically the same for “forward” and “inverse” problems. However, “forward” and “inverse” problems come with their own peculiar challenges: For example, ill-posedness is often a major concern for “inverse” problems (that means multiple valid solutions could explain a given observation) while speed and accuracy are of particular importance in “forward” simulations. These differences are also reflected in the different approaches to scale invariance (Holl et al. use a hybrid approach that makes use of embedding updates from a scale-invariant inverse problem solver to improve a gradient-descent based learning pipeline for inverse problems while our approach makes use of an internal state variable to automatically scale update steps for forward simulations).

---

### Official Review · Reviewer_gC5k · 2024-10-30

**Soundness:** 1
**Presentation:** 2
**Contribution:** 2
**Rating:** 3
**Confidence:** 3

**Summary:**

The paper presents a neural-network-based meta-optimizer, Metamizer, designed to minimize the residuals of physical systems, especially partial differential equations (PDEs), in pixel space. The authors conduct experiments comparing the performance of Metamizer to both gradient-base optimzers and linear system solvers. Metamizer is evaluated on different PDEs and the results presented.

**Strengths:**

The problem the paper tries to address is of interest and has wide applicability.

**Weaknesses:**

The paper has 3 main weaknesses:
- **MW1**: Lack of quantitative evaluation. Only qualitative images and residuals are presented, making it difficult to assess the method's performance objectively. This issue affects all the experiments.
- **MW2**:  No baseline comparisons with other Learning to Optimize (L2O) methods, classical numerical solvers, or neural PDE methods.
- **MW3**:  Limited experimental scope. Experiments use only fixed-size grids, 100x100 (331) and 400x400 (775), which restricts generalizability and does not test varying resolutions, where new dynamics may arise, especially in the Navier-Stokes equation.


## Main weaknesses

The proposed method can be viewed as a meta-learning method, a PDE solver, or an emulator but it lacks comparisons with other methods from these fields. Ideally, the method would be compared against methods from each of these fields. I acknowledge that such an exhaustive comparison involves a considerable effort, but the paper omits even a basic evaluation against standard methods in these areas.

The authors assert that L2O has not been applied to physics (99) but fail to demonstrate how Metamizer performs in the physics context against other L2O methods. Furthermore, the paper’s accuracy claims lack quantitative evidence, as it provides only qualitative images and residuals. This is inadequate for judging solution quality, especially when standard neural PDE literature benchmarks against ground truth obtained from numerical methods. Moreover, the lack of comparison to classical numerical methods is striking, since a comparison is performed against linear solvers for a linear system built in pixel space. Effectively, the paper is comparing against different linear solvers for the finite element method.

In short, the experimental evidence does not convincingly support the method's efficacy or comparative performance and extensive existing literature has not been engaged with.

## Additional Weaknesses

**W1**: The claim that no hyperparameters are used (68-69) is inaccurate, as $s_0$ is a hyperparameter (238).

**W2**: The claim on (332-333) is misleading. In the deep learning literature, accuracy typically reflects error metrics like MSE between the ground truth and the prediction. In this paper, accuracy has a different meaning (value of the residual) so it is not directly comparable. This is related to **MW1-2**.

**W2**: The accuracy claim on (332-333) is misleading; in deep learning, accuracy typically reflects error metrics like MSE. It is not possible to directly compare it to the claimed MSR.

**W3**: Misleading claim about linear solvers and their usage. In Fig. 6 Metamizer is compared to linear solvers, and in (327) it is claimed that linear solvers are limited to linear PDEs. This is only true in pixel space. Most PDE-solving techniques, such as the FEM, rely on solving a linear system derived from the equation through linearization [1]. Related to **MW2**.

**W4**: The caption of Fig. 8 claims that Metamizer can simulate a wide range of Reynolds numbers and shows snapshots for $Re=1$, $Re=50$ and $Re=200$. This range is insufficient compared to the typical Reynolds number range (1 to millions) in fluid dynamics, missing the complex behaviors at higher Reynolds numbers [2].

**W5**: No limitations whatsoever are discussed or acknowledged.


-------
Given all of the above, I recommend the paper be rejected.

**Questions:**

## Missing details/questions

**Q1**: Insufficient details on residual computation (148-154, 175), specifically handling spatial resolution and Navier-Stokes pressure. More information should be provided in the appendix (related to **MW3**).

**Q2**: In (198-211) you say that the main problem with directly optimizing the gradient is the scaling, therefore you introduce a scale-invariant optimizer. What about local minima? The solution of the physical problem is the global minimum of the problem, so getting stuck in a local minimum is a potential issue that is not necessarily solved by scale invariance.

**Q3**: Is there no cold start issue? In the **Training** paragraph (256-269) it is stated that the training pool is initialized with randomized solutions. How exactly?. Fig. 6 shows that the gradient-based optimizers (Adam, SGD, etc) fail to converge. If the gradient signal is so poor how can the UNet learn from it? How does this relate to the scale vs local minima issues from Q1? Can Metamizer, once trained, minimize for randomized solutions drawn from a different distribution than the one it was trained on?

**Q4**: I appreciate the wall time and GPU usage training information (269). How many update steps were performed on the model? How many data samples did it see (presumably the number of update steps * batch size)?

**Q5** I find the description of the time integration and how it is used in the context of Metamizer unclear. Is the integration scheme used to define the discretization used of the time derivative? E.g. for forward Euler it would be approximated as $u_t' = \frac{u_{t+1} - u_{t}}{\Delta t}$

**Q6** Why is time integration needed at all? Can the residual computed on the full trajectory not be minimized for all the states at once?


## Things to improve the paper that did **not** impact the score:

Abstracts usually have a single paragraph.

Space before comma on line 93.


## Refrences

1) Brenner, S. C., & Scott, L. R. (2008). The Mathematical Theory of Finite Element Methods. In Texts in Applied Mathematics. Springer New York. https://doi.org/10.1007/978-0-387-75934-0
2) Chassaing, P. (2022). Fundamentals of Fluid Mechanics. Springer International Publishing. https://doi.org/10.1007/978-3-031-10086-4


## Edit history
1. Confidence reduced from 4 to 3 in light of the discussion with the authors.

---

> ### Author Response · Authors · 2024-11-19
>
> Dear reviewer,
> Thanks for your remarks and questions!
>
> Regarding:
>
> **Main Weakness 1:** Presenting the residuals of equations is a very common form of quantitative evaluation in the field of physics-based DL. Presenting MSE to ground truth values as suggested in Weakness 2 is often not possible since chaotic systems (such as for example Burgers-Equation / Cloth simulation / Navier-Stokes equation at high Re) diverge after small initial perturbations.
>
> **Main Weakness 2:** We compared our method to several classical solvers such as gcrotmk, minres, gmres, lgmres for the finite difference scheme. Furthermore, we added GPU baselines (see Section 1 in Rebuttal.pdf of Supplementary). Of course, countless further schemes exist such as FEM, FVM, lattice Boltzmann methods etc, however, in this work we focus on FD as a popular approach on regular grids. So far, L2O methods have not been applied to develop a method that generalizes across multiple different PDEs, thus, a meaningful comparison was not possible. Other neural PDE methods only focus on single PDEs and/or report orders magnitudes less accuracy. Thus, a fair comparison is not possible.
>
> **Main Weakness 3:** We present results for 7 different PDEs that were generated by a single neural network. There are no other methods that allow a single neural network to handle such a wide range of PDEs. Furthermore, PDEs can be scaled fairly easily to fit the resolution of the network. Figures 7,8,9,10,11 show clearly that Metamizer can handle a wide range of different dynamics that occur at various scales (Figure 8 focuses on the mentioned Navier-Stokes equation).
>
> **Weakness 1:** $s_0$ is not really a hyperparameter as it was kept the same across all different PDEs (even those that were not considered during training).
>
> **Weakness 2 (and it’s repetition):** See our reply to main weakness 1.
>
> **Weakness 3:** In contrast to linear system solvers, Metamizer can be directly applied to non-linear PDEs without requiring any linearization tricks that might introduce inaccuracies.
>
> **Weakness 4:** We only went to $Re=200$ since higher Reynolds numbers would result in high frequency turbulences in the velocity and pressure field that can not be properly resolved on a 100x100 grid without additional regularization techniques. Nevertheless, Metamizer captures many complex fluid behaviors such as the Bernoulli effect or von Karman vortex streets.
>
> **Weakness 5:** So far, we did not consider Neumann or Robin boundary conditions. Furthermore, our network only works on a grid data structure and doesn't consider coupled PDEs such as the coupled Navier-Stokes and Advection-Diffusion equation. We addressed these limitations in our conclusion section (albeit we tried to formulate these limitations in a constructive way).
>
> **Question 1:** The residuals were computed with standard finite differences and we used the units of the grid (one grid cell corresponds to a unit length “1”). We’ll elaborate on that in the appendix.
>
> **Question 2:** Local minima didn't cause any problems in our experiments. We suppose that local minima occur only very rarely in physics-based losses (for example if you would have a meta-stable “clicker” state in the internal energy function of cloth). For PDEs such as the Laplace or Poisson equation there exist no other local minima besides the global minimum.
>
> **Question(s) 3:**
> *Is there no cold start issue?*
> Yes, it takes a bit of time at the beginning to fill up the training pool with realistic training data. However, this is not a big issue.
>
> *In the Training paragraph (256-269) it is stated that the training pool is initialized with randomized solutions. How exactly?*
> The training pool is not initialized with randomized solutions (this would be computationally expensive) but only with randomized boundary / initial conditions that can be generated on the fly. This involves for example randomly moving boxes in the fluid domain or random fix-points for the cloth. We'll add a more detailed description in our appendix.
>
> *(Adam, SGD, etc) fail to converge...*
> Actually, Adam, SGD etc do not really fail to converge but find quite plausible solutions. However, they plateau out fairly quickly at around 10^-6. We also tested smaller learning rates, however this resulted in slower convergence and did not significantly reduce the residual errors (we added a comparison of various learning rates in Section 4 of Rebuttal.pdf).
>
> *If the gradient signal is so poor how can the UNet learn from it?*
> The U-Net can learn from these very small gradients because we use gradient normalization, normalize the inputs and use the described scaling mechanism to allow for scale invariant update step sizes.
>
> *Can Metamizer, once trained, minimize for randomized solutions drawn from a different distribution than the one it was trained on?*
> Metamizer was able to generalize to the Poisson, wave and Burgers equations although these equations were not covered during training.

---

> > ### Author Response · Authors · 2024-11-19
> >
> > **Question 4:** We trained our network for roughly 50000 update steps with a batch size of 10. So Metamizer saw about 500000 samples during training.
> >
> > **Question 5:** Exactly. For stability reasons, we mainly used backward Euler and the Crank-Nicolson scheme.
> >
> > **Question 6:** Yes, time integration is needed for time-dependent PDEs. Minimizing later PDE states without having solved their predecessor states would waste lots of computational resources.
> >
> > **Further remarks:** Thanks for pointing out the typo. Furthermore, we’ll try to condense the abstract a bit.

---

> ### Comment · Reviewer_gC5k · 2024-11-21
>
> **Main Weakness 1**: The chaotic nature does not justify not presenting a comparison with a baseline (i.e. MSE to ground truth), as chaos takes time to form, a period where the trajectories are expected to be similar can be shown. This is done in both Neural Operator (NO) and autoregressive models (AR) literature, and I don't see why it should not be done here. Moreover, as you point out, only Burgers-Equation, Cloth simulation, and Navier-Stokes present chaotic behavior. As for your claim that the residual is common, it is partially true. It is usually shown together with a ground truth as in Raissi et al. (2017) and many others.
>
> **Main Weakness 2**: I acknowledge your comparison to methods that work on regular grids. However, schemes like FEM and FVM are standard and SOTA, and therefore cannot simply be ignored. Similarly, the fact that you claim that L2) has not been applied to PDEs (some reviewers disagree with this claim) does not justify not comparing against them. As you point out in another answer, those L2O methods are general, and therefore can be applied here. How do we know Metamizer is better than any generic L2O with a PDE-based loss?
>
> *Main Weakness 3*: What do you mean the PDE can be scaled to fit the resolution of the network? The PDE has an infinite resolution, and the discretizations will have finite resolution. Depending on that resolution, the solutions to the (restricted) PDE can be dramatically different. It does not follow from the method working at a given resolution that it will generalize.
>
>
> **W1**: I disagree, it is a hyperparameter to which you have given a specific value. It is like saying that the learning rate is not a hyperparameter because you have used the same value during all the experiments.
>
> **W2**: as above
>
> **W3**: Yet there is no comparison to solutions provided by solvers that make use of linearization, so it is impossible to judge whether this claimed advantage materializes.
>
> **W4**: The high frequency turbulence is precisely the challenging aspect of the NS equation, and it cannot claimed to be solved effectively if this is not shown. What about the 400x400 grid, did you try NS on that resolution with a higher Re? The need of additional regularization is what I had in mind in *MW3* and *Q1*. My concern is, will Metamizer work outside of very simple dynamics?
>
> **W5**: The need for further treatment of NS at high Re is not addressed, nor the lack of testing of challenging, high resolution dynamics.
>
> **Q1,4,5,6**: Thanks for the clarification.
>
> **Q2**: Thank you, *Q3* clarified this for me.
>
> **Q3**: Thanks for the clarification. I had slightly misunderstood your formulation. In light of this you should consider TENG: Time-Evolving Natural Gradient for Solving PDEs (...) by Chen et al. (2024, ICML). While they do not use L2O, they also claim machine precision.
>
>
> As my main concerns remain I will keep my score.

---

> ### Author Response · Authors · 2024-11-22
>
> Dear reviewer,
> Thanks for engaging in the discussion!
> We are happy that we could clarify your questions!
>
> Regarding:
>
> **Main Weakness 1**: Now, we added a MSE comparison of Metamizer and various numerical solvers to the analytical solution of the Laplace equation in Section 5 of the Rebuttal. As expected, this error correlates very much with the residual errors provided already. The MSE is often reported in data-driven deep learning (for example in case of NO or AR models applied on weather data where residual errors cannot be computed). For physics-driven approaches, the underlying residual errors already give a very good hint on how well PDE solutions are approximated.
>
> **Main Weakness 2**: We’ll add more discussion of different numerical schemes (FEM/FVM) in our related work section. However, an extensive comparison of such schemes is not the goal of this work as there is already plentiful literature on that matter (see e.g. “Finite difference, finite element and finite volume methods for partial differential equations” by Peiro et al. (2005)). Extending our method to support a loss based on the Galerkin method similar to “Galerkin Neural Networks: A Framework for Approximating Variational Equations with Error Control” by Ainsworth et al. (2021) could be an interesting direction for future research. It's worth noting that other Learning to Optimize (L2O) papers don't seem to report the same level of accuracy in their results as Metamizer. We believe this is mainly due to the fact that these methods have been taylored for very different contexts (e.g. optimizing neural nets) and don't have a scaling mechanism.
>
> **Main Weakness 3**: Let’s clarify that: A PDE on a small 1mm x 1mm fluid domain can be scaled onto a 100x100 grid by adjusting $\mu$ and $\rho$ accordingly. If e.g. $Re=1$, the solution will look similar to Figure 8a. On a bigger 20cm x 20cm domain, the Reynolds number might be 200 and if you scale that PDE to a 100x100 grid, the results will look similar to Figure 8c. Scaling inputs to fit the resolution of a CNN is common practice in DL so we do not consider this a severe weakness of our work. Of course, if you have a fluid with even larger Re numbers that can not be properly resolved on a 100x100 grid due to high frequency turbulences, you indeed have to go to larger grid sizes. We’ll address this in our conclusion section. In the meantime, we tested our approach for the Navier-Stokes equation on a 400x400 grid and obtained complex solutions for turbulent fluid dynamics at $Re=2000$ (see Section 6 in Rebuttal).
>
> **W1**: Usually, I wouldn't call a constant value that doesn't require any tuning a hyperparameter but maybe there are different opinions. To avoid misunderstandings, we'll change the wording to "Metamizer doesn't require any tuning".
>
> **W3**: Directly minimizing the residuals of nonlinear PDEs is one of the key objectives of countless physics-based DL approaches. “High Order Numerical Solutions to Convection Diffusion Equations with Different Approaches” by Liu et al. (2015) compares various approaches with and without linearization for Burgers equation and justifies direct nonlinear optimization to avoid truncation errors.
>
> **W4**: We added further results for the Navier-Stokes equation at Re=2000 on a 400x400 grid in section 6 of the Rebuttal. As can be seen in Figure 8, Metamizer can handle complex turbulent fluid dynamics.
>
> **W5**: We’ll discuss the current limitation of a fixed resolution grid in Section 6 and add a remark that high Reynolds numbers would require retraining on a larger grid.
>
> **Q3**: Thanks for the interesting reference. Indeed TENG achieves high accuracy with implicit PINNs. However, this method requires retraining the neural net for every PDE which takes between $10^3-10^5$ seconds. Our method doesn’t require retraining and solves PDEs in around $10^{-1}$ seconds.

---

> > ### Comment · Reviewer_gC5k · 2024-11-26
> >
> > **MW1**: Thank you for the additional comparison. However, due to the machine learning nature of the method and lack of theoretical guarantees, the small residual alone is insufficient to assess solution quality. For instance, PINNs can also achieve small residuals but produce poor solutions. Further, residuals are problem-dependent, so additional context (e.g., comparison to ground truth) is important. While you provide ground truth for the Laplace equation, this does not generalize to other experiments. I again remark that other physcis-driven approaches, such as the original PINN from Raissi et al. (2017) report the ground truth together with the residuals.
> >
> > **Main Weakness 2**: I understand, and agree, that an extensive comparison with all equation solving methods is not the point nor feasible. My point is that just finite difference linear solves is not enough, as you have non linear equations that are routinely solved with these methods (e.g. FVM or pseudospectral of NS) and claim Metamizer will outperform solvers on them too. This is also related to **MW1**, as the lack of comparison means you don't have a ground truth to present together with the solution so that its quality can assessed. As for L2O, if you claim they don't report the same level of accuracy on their bechmarks as Metamizer, then why not show it? How can we know that their performance in this case will be inferior to Metamizer? Will any black-box L2O method work with how your training methodology?
> >
> > **MW3**: Thank you for adding NS results; it is a step forward. However, my concern centers on performance at resolutions that introduce previously unseen dynamics. Rescaling inputs alone may not address this. Comparisons to proven solvers under such conditions would help validate solution quality. This is analogous to verifying generative models, where plausibility alone is insufficient, and quantitative benchmarks are essential.
> >
> > **W1**: The phrasing around hyperparameter tuning could be clearer. For example: "We found a constant value for Metamizer's single hyperparameter that performed well across all experiments." This suggests empirical robustness but avoids implying zero tuning effort. We don't know if it doesn't require tuning, we know that you happened on an effective value (maybe on the first try) and kept it constant, there might be better values or there might be not.
> >
> > **W3**: I understand the advantages of directly optimizing the nonlinear residual. I'm saying that without any information besides the residual value it is hard to judge the quality of the solution. For example, will there be any error introduced by Metamizer to offset the gains made by directly optimizing the nonlinear residual?
> >
> > **W4**: This relates directly to **MW1** and **MW3**. Comparisons to trusted solvers remain critical to evaluate quality across experiments.
> >
> > **W5**:  Thank you for addressing this. The remark should not be about high Re in particular, but rather general about unseen dynamics during training.

---

> ### Author Response · Authors · 2024-11-29
>
> Dear reviewer,
> Thanks for your reply!
>
> Regarding:
>
> **MW1:** Actually, quite a lot of theoretical work exists on a posteriori error estimators based on the residuals for finite differences and, more recently, also for PINNs. See for example:
> - “A posteriori error estimates in finite difference techniques” by Kelly et al. (1987)
> - “Finite Difference Methods and Spatial A Posteriori Error Estimates for Solving Parabolic Equations in Three Space Dimensions on Grids with Irregular Nodes” by Moore (1999)
> - “A posteriori error estimates for fully discrete finite difference method for linear parabolic equations” by Mao et al. (2024)
> - “Error estimates for physics informed neural networks approximating the Navier-Stokes equations” by Ryck et al. (2023)
> - “Solving PDEs by variational physics-informed neural networks: an a posteriori error analysis” by Berrone et al. (2022)
>
> Also, numerous important works in the field of physics-based deep-learning nowadays rely solely on residual based accuracy evaluations. See for example:
> - “Accelerating Eulerian Fluid Simulation With Convolutional Networks” by Tompson et al. (2016)
> - “A deep conjugate direction method for iteratively solving linear systems” by Kaneda et al. (2023)
> - “A Neural-preconditioned Poisson Solver for Mixed Dirichlet and Neumann Boundary Conditions” by Lan et al. (2023)
>
> The work by Raissi et al (2017) is different from our approach since they train an implicit neural representation on a physics-based loss but also on sparsely sampled observation data. In this case, a comparison to the “ground truth” of the observed data seems to make sense.
>
> **MW2:** We do not compare against or make claims about outperforming different numerical schemes such as FEM or FVM etc. Instead, our goal is to develop an efficient and versatile solver that can solve a wide range of linear / nonlinear systems of equations given by FD and evaluate the performance based on their residuals. Experimenting with other numerical schemes such as FEM or FVM would be orthogonal to our work and could be an interesting direction for future research (we included a corresponding remark about Galerkin neural networks in our conclusion section).
> Regarding other L2O approaches: During the initial development of Metamizer, we did not have a scaling mechanism or gradient normalization and did not obtain good results at all. Thus, in our opinion, comparing against other L2O approaches that are specialized in different domains (e.g. in optimizing the weights of neural networks) and do not use a scaling mechanism / gradient normalization / U-Net / etc would have been unfair (or even infeasible for approaches that rely on different underlying data structures). Thus, we found a comparison to numerical solvers (CG, GMRES etc) more appropriate.
>
> **MW3:** The turbulent chaotic flow fields shown in Appendix A were clearly not seen during training. Furthermore, Metamizer generalizes to new dynamics of PDEs that were not covered during training such as the wave and Burgers equation. As discussed above, we validated our model quantitatively based on the residuals and got machine precision level accuracy for the Laplace, advection-diffusion, wave and Burgers equation.
>
> **W1:** Thanks for the remark. We changed the wording in the updated submission and hope this is clear now.
>
> **W3:** The residual errors reported in Table 1 already contain all errors that might be introduced by Metamizer.
>
> **W4 / W5:** See previous replies to MW1 / MW3.

---

> > ### Comment · Reviewer_gC5k · 2024-11-29
> >
> > **MW1**: Thank you for the numerous references. I will peruse them as I'm not familiar with them. In my experience the practical totality of the papers report ground truths for comparison, even when not using observations, and it seems necessary and natural to me. I do not know the quality of a result without a reference solution (for example, real experiments to verify against). However, I will reduce the confidence of my review to reflect the fact that I'm not familiar with the cited references.
> >
> > **MW2**: For the L2O, I think it would have been a fair comparison. They do not claim to be optimal for physics based losses, and as long as you made that clear you can use them as baselines. Those experiments would have been very informative. Again, right now I have no idea of how Metamizer compares to other L2O techniques. This comparison is not meaningless as Metamizer is undoubtedly a meta learning approach, and you acknowledge it as such.
> >
> > Regarding FEM and FVM, I would have liked at least one of them in the comparison in Fig. 6, and for Fig. 6 to include other equations, not just the Laplace equation, which is the simplest of all the tested equations. As it stands now it is a limited benchmark.
> >
> > **MW3**: I can see the new dynamics in Fig. 13. However, just 4 snapshots without any further information is not convincing evidence. MW3 was about this type of generalization, not one to different equations which you defend (I still have my doubts about the validity of the results in general from **MW1**).
> >
> > **W1**: Thanks, **W3**: Good point.
> >
> > In summary, thanks for addressing the minor weaknesses and answering the questions. I'm not convinced about the major weaknesses for the reasons above, mainly the reference solutions in **MW1** and the lack of L2O in **MW2**. However, as explained in **MW1**, I reduced my confidence to reflect my unfamiliarity with the cited literature.

---

### Official Review · Reviewer_EsdF · 2024-11-04

**Soundness:** 2
**Presentation:** 2
**Contribution:** 3
**Rating:** 5
**Confidence:** 4

**Summary:**

The paper proposes a meta-optimization technique for solving physics simulation problems, including linear and nonlinear PDE problems.  The proposed Metamizer framework is an iterative optimizer that uses a scale-invariant architecture to improve gradient descent updates to accelerate convergence.

**Strengths:**

1. A nice advantage of Metamizer is that it's able to be applied to PDEs that weren't seen during training.  I don't know if there are other state-of-the-art methods like that, but I appreciated that feature.

2. Metamizer appears to be scale-invariant and to not need retraining for different PDEs, which again, is quite useful from a practical perspective.

3. Metamizer seems like a simple idea with a relatively straightforward architecture, which means that future research in this direction should be possible to build upon the present work.

**Weaknesses:**

1. "However, to the best of our knowledge, L2O has not yet been applied in the context of physics simulations"
Unfortunately, I must inform the authors that L2O has absolutely been applied in the context of physics simulations.  Please see for example the works "A deep conjugate direction method for iteratively solving linear systems" (ICML 2023) and "A Neural-Preconditioned Poisson Solver for Mixed Dirichlet and Neumann Boundary Conditions" (ICML 2024) and the references / related work discussions in there.

2. "Data-driven Deep Learning approaches"
The authors may be interested recent works like "Toward Improving Boussinesq Flow Simulations by Learning with Compressible Flow" (PASC 2024), which show that the addition of a neural surrogate can produce results that are more accurate than what is possible with a classical numerical solver - because a solver may be using a less accurate physical model.  This is an important point to be aware of or mention, since many deep learning approaches overlook the fact that one way to improve efficiency and accuracy is to simulate things with a cheaper model and then use learning models trained on more accurate models.

2. "Unfortunately, the learned implicit representations need to be retrained for every physical simulation and thus are not real-time capable."
This criticism of PINNs does not make sense to me.  When you train a PINN, you can evaluate is very quickly.  The fact that PINNs are best when trained on a specific physics problem, does not seem correlated to the performance of evaluating (performing inference on) a neural network like a PINN.  I suggest the authors remove this criticism or clarify what they mean.

3. (Minor) In the section "Time-Integration Schemes," it is likely worth tossing in a reference to a textbook or some similar resource that discusses additional common time integration schemes like RK4 - just to avoid giving the impression that the three schemes mentioned are the only ones.

4. It is interesting the authors' technique seems to essentially just be a U-net with carefully-chosen features (Fig. 3).  With physics problems, solutions are often smooth/continuous, so practitioners often use CNNs or perhaps mix convolutional layers with U-nets.  There is no discussion of why a plain U-net architecture was chosen, and adding something like that would be helpful for readers to gain intuition into the method.

5. There's not much that can be gleaned from Figs. 7 or 8 without quantitative metrics or labeled error/scale bars.

6. L757 "This is because Metamizer can be effectively parallelized on a GPU." Sparse iterative linear solvers can be effectively parallelized on a GPU - in fact, this is one of the things GPUs are best at.  This statement makes me believe the authors are using CPU or perhaps even serial implementations of things like CG or MINRES, which is a severely (10x+) unfair comparison.

Minor:

7. L245 you call it a CNN but it's specifically a U-Net?

8. L264 You say you use gradient descent but in parentheses you say Adam - do the authors understand these are distinct numerical methods?

9. L268 "ca 6 hours" should just be "about 6 hours," or even better, just specify the time more precisely, e.g., "6 hours and 15 minutes"

10. The abstract is a little long and split into a few paragraphs; I would recommend condensing things a bit.

11. Formatting mistakes like not doing LaTeX quotes properly - the authors should know better than this in a venue like ICLR

12. L235 citation not formatted right (\citet not \citep)

13. L277-281 (for instance - there are other places for this) Put commas after e.g., this is a grammar thing

14. L434 probably a typo in the table here

**Questions:**

1. The authors criticize gradient descent-like methods like AdaGrad and Adam, yet their method is based on improving gradient descent.  Can the authors speculate on how their ideas could work if based on other techniques like Adam?

2. L251-254 The authors describe how they want to use one network for all PDEs but then describe how different PDEs require different numbers of channels.  This sounds like different networks need to be trained depending on how many variables are in the problem (e.g., you'd have a 1-variable network, a 3-variable network, etc.).  Can the authors clarify this?

3. Figure 6 and the discussion around it doesn't say what equation and what boundary conditions are being solved here.  If this is Laplace/Poisson (at least without all-Neumann boundary conditions) - I don't really believe the results, because this should be a symmetric positive-definite sparse linear system that conjugate gradients should perform quite well on (and certainly better than GMRES and other slower/more general Krylov solvers).

4. With regards to the Figure 6 results, it's a little unfair to compare against constant-learning-rate variants of the other methods shown.  Of course things like SGD can't converge to 1e-33 precision with a learning rate of 0.01 - it is probably doing a very good job but the learning rate is so large that it literally cannot land on the minimum.  And it's not a mystery how to choose step sizes that enable convergence - e.g., for gradient descent, the Wolfe conditions (for instance) can inform what step size you should take at each step.

5. These results are further unfair because Metamizer - along with any other L2O methods used for these types of problems, see for instance the discussion in Kaneda et al. 2023 - are effectively acting as preconditioned solvers.  In Metamizer's case you're preconditioning gradient descent by using data about PDE losses.  So it would be fairer to compare against preconditioned CG, MINRES, GMRES, etc.  Unfortunately, the timings and iteration counts are so close that I bet if you used even just a Jacobi/diagonal preconditioner on those methods, they would all be outperforming Metamizer.

6. Figure 4 indicates that Metamizer-produced solutions may not enforce much smoothness (e.g., there is some rough noise around 20 iterations, where solutions are still not in the regime of double precision limits).  As mentioned earlier, I might venture a guess that this is due to the U-Net architecture chosen.  However, I am curious for the authors' thoughts?  This may be a useful thing to discuss in the paper, too.

Although my score leans somewhat negative (I would prefer to mark this as a 4, but it's not an option), it is mostly due to the evaluation and presentation, and I believe this is a worthwhile idea that could be published in an appropriate venue and with suitable manuscript improvements.

---

> ### Author Response · Authors · 2024-11-19
>
> Dear reviewer,
> Thanks for your thorough remarks and questions!
>
> Regarding:
>
> **Weakness 1:** Thanks for pointing out these additional references that show impressive results for the Poisson Equation by improving conjugate gradients with a neural preconditioner. However, both of these methods did not go beyond the Poisson Equation in the specific context of a pressure projection step for incompressible fluids. The terms “Meta-learning” and its subfield “L2O” usually refer to more general purpose optimizers and weren’t mentioned in these works (although learning a preconditioner for CG could be also considered a form of L2O). Our approach is kept more general and allows a single neural network to solve a wide range of PDEs, including elliptic, parabolic and hyperbolic PDEs as well as nonlinear PDEs without retraining. In contrast to neural preconditioned CG methods, our approach doesn’t require access to the underlying matrix $A$, $A$ to be symmetric or even residuals (as shown in our cloth simulation examples) but only gradients. Furthermore, the mentioned works demonstrate a residual norm reduction by around 4-6 orders of magnitudes while our approach is able to reduce the residuals up to machine precision (>10 orders of magnitudes). On top of that, both of these methods require fairly expensive training data generation while our method doesn’t require any precomputed training data at all. We’ll add a corresponding discussion in our updated related work section.
>
> **Weakness 2:** This is not a particular weakness of our approach but a general point that differentiates data-driven deep learning from physics-driven deep learning (as well as numerical solvers): If the underlying physical model is not fully known or the resolution of the domain discretization does not allow to solve the equation, you should go with data-driven approaches. However, this requires lots of high quality training data. But if the physical model is known and enough resources for proper domain resolution are available, you can achieve much higher accuracy with numerical and physics-driven approaches - even if no training data is available at all. Here, of course we assume the latter scenario.
>
> **Weakness 3:** Let us clarify: Yes, implicit PINNs can be evaluated very quickly once they are trained on a specific physics problem. However, a PINN cannot be evaluated for example on the wave equation after it was trained on the Laplace equation. In contrast, Metamizer can be directly applied to various PDEs with different BCs and ICs without retraining. This allows for interactive real-time applications that are impossible with implicit PINNs.
>
> **Weakness 4:** Thanks! Indeed, we only mentioned the 3 most popular time-integration schemes but there are also a lot more schemes that could be used. We’ll add a corresponding reference.
>
> **Weakness 5:** The U-Net architecture has been successfully applied in lots of physics-related works. This is because it allows to deal with long-range dependencies at coarse resolutions while preserving small details at fine resolutions. In PDEs, the domain of dependence can be relatively small (e.g. in hyperbolic PDEs) but also span the entire domain (e.g. in elliptic PDEs). We’ll add a corresponding discussion to motivate our architectural choice.
>
> **Weakness 6:** Indeed, Figures 7 & 8 present qualitative results. For quantitative results, we refer to Table 1.
>
> **Weakness 7:** Indeed, these GPU baselines were missing in the current version. Thus, we now added further comparisons to GPU based sparse linear system solvers (including CG,GMRES,MINRES) using the CuPy package (see Section 1 in Rebuttal.pdf of our Supplementary). This indeed significantly improved the baselines on the 400x400 grid, but Metamizer still remained faster.
>
> **Minor 7:** Yes, we meant the U-Net as a special form of a CNN.
>
> **Minor 8:** Yes, we understand that Adam is one of many stochastic gradient descent derivatives. To make it clearer, we’ll change that to "optimize the Metamizer with Adam (lr=0.001)"
>
> **Minor 9 - 14:** Thanks for the remarks. We'll fix that!
>
> **Question 1:** Incorporating ideas from Adam / AdaGrad such as keeping track of higher order momenta of the gradients to estimate their variance could be an interesting direction for future research.
>
> **Question 2:** We only trained one single network that can solve all of the mentioned PDEs (we consider this the main contribution of our work). Because different PDEs can have different numbers of variables (channels), Metamizer optimizes every variable individually.
>
> **Question 3:** Figure 6 shows results for the Laplace Equation. We'll clarify this also in the figure caption. Indeed, CG didn't converge since the matrix wasn't symmetric due to our naive implementation of the Dirichlet boundary conditions. We already fixed that (see Section 2 in Rebuttal.pdf). While on small grids (100x100), this led to faster convergence, Metamizer is still far ahead on larger grids (400x400).

---

> > ### Author Response · Authors · 2024-11-19
> >
> > **Question 4:** For SGD, Adam, etc we actually tested various different learning rates (also much smaller ones) but only observed slower convergence without significant improvements with respect to the final residual errors. We added supplementary plots with different learning rates (see Section 4 in Rebuttal.pdf). Figure 6 only reports the learning rates with the best trade-offs between convergence speed and accuracy.
> >
> > **Question 5:** The Jacobi-precoditioner, defined as diag(A), is basically just the identity matrix in case of the finite difference Laplace operator and thus wouldn’t help in our case.
> > Furthermore, as already mentioned in our reply to Weakness 1, Metamizer does not act as a neural preconditioner for CG but is a much more general optimizer.
> >
> > **Question 6:** This noise is not in the solution but in the update steps (scaled by 1e-7) and gets ironed out by Metamizer after a few more iterations (as can be seen in our supplementary video). We’ll add a corresponding discussion.

---

> > > ### Comment · Reviewer_EsdF · 2024-11-24
> > >
> > > W1: Please note the Kaneda and Lan works are able to reduce the residual by any desired amount; they can reach arbitrarily low tolerances.  The authors in those two papers simply set a tolerance of around 10^-4 for their examples, but the key advantage of their method over Tompson et al. and similar works is that their techniques allow arbitrary residual reduction, just like a standard CG solver.  The other aspects of the authors' comments here indeed seem like attractive features of Metamizer.
> > >
> > > - In the rebuttal PDF (which I hope makes it to the appendix or main paper?), it would be more appropriate to call methods like GMRES, CG, etc. "Krylov subspace methods" than "conjugate gradient methods."  Also, GMRES does not require A to be symmetric (that's the whole point of GMRES over MINRES).
> > >
> > > - In section 2 of the rebuttal, I don't really understand this comparison.  Why is it fair to compare your GPU implementation to a CPU solver?  You claim Metamizer is 10x faster than a CPU CG solver.  But I have implemented CG on CPU and GPU, and I know first-hand that CG runs about 10x faster on a GPU.  So I guess I'm not sure what you're trying to show here.  If you wanted to run Metamizer on the CPU for the examples here, I think that would make this a lot more useful.  Open to the authors' thoughts.
> > >
> > > - Rebuttal Fig 7a, why do some of the curves not start at t=0?  Anyway, I'm not convinced this section / this figure is necessary, given Fig. 2.  Other reviewers can chime in here but I feel that Fig. 2 captures already what is being shown here.
> > >
> > > - Regarding Section 1 and Figs 1/2 (and to some extent Section 2 and Figs 3/4, though see above): it looks like you are using non-preconditioned / vanilla versions of these solvers.  While these are acceptable baselines, in practice, anyone using a Krylov solver will apply preconditioning - at the very least diagonal preconditioning, and in the case of something like CG, incomplete LU or incomplete Cholesky preconditioners, or algebraic multigrid preconditioners (which are theoretically optimal in terms of reducing number of iterations - though in terms of solution time, Jacobi/diagonal or ILU/IC can be faster).  So it's a little disingenuous that the authors are only showing that Metamizer outperforms the vanilla variants of methods that no one would actually use in practice.  Given that even a diagonal preconditioner can speed up Krylov solvers by like 10x, I'm curious to see how Metamizer compares against preconditioned solvers.  If Metamizer is to make a "profound impact" on numerical solvers as the authors claim, it should be able to beat these standard preconditioned methods that are what people actually use.
> > >
> > > Given that the authors have addressed the majority of my questions and concerns, I am going to increase my score.  Nonetheless, I think that there are still some weaknesses with the present submission as mentioned here.  If the authors upload a revised version of their manuscript (per ICLR policy) before the author-reviewer discussion period ends - a version that addresses all the feedback and proposed changes so far - I will review it and reconsider my score, including in light of any improvements the authors may make thanks to the other reviewers.

---

> > > > ### Author Response · Authors · 2024-11-26
> > > >
> > > > Dear reviewer,
> > > >
> > > > Thanks a lot for your reply and acknowledging the improvements of our submission!
> > > >
> > > > Regarding W1: Thanks for the clarification! We’ll weaken our claim about “unprecedented accuracy for deep-learning based approaches” and instead focus more on Metamizers ability to achieve machine precision across multiple PDEs without retraining.
> > > >
> > > > Regarding your remarks:
> > > > - Yes, we plan to include all of the results of the Rebuttal in the main paper or appendix of our updated submission. Thanks for the remark about Krylov subspace methods, we’ll clarify that.
> > > > - We kept the CPU comparison because we realized that on a 100x100 grid, the CPU implementation was actually faster than the GPU implementation. However, on a larger 400x400 grid, the GPU implementation significantly outperforms the CPU. Our plan is to replace Figure 6 of the main paper (performance comparison to scipy on the CPU) with Figure 1 of the Rebuttal (performance comparison to CuPy on the GPU) and discuss the different problem size dependent scaling behaviours for CPUs and GPUs in the appendix.
> > > > - The curves in Figure 7a start after one iteration which can take a certain amount of time for different methods. Since the time axis is logarithmic, unfortunately, we cannot display t=0. We included this Figure since Reviewer gC5k asked for a ground truth comparison and we believe this comparison could help readers that come from a more data-driven deep-learning background.
> > > > - Yes, we reported non-preconditioned results in these Figures. In the meantime, we tested preconditioners for conjugated gradients based on the incomplete LU factorization and the algebraic multigrid method using pyAMG (see Section 7.1 in Rebuttal.pdf). In particular, AMG showed significantly improved convergence speed. Unfortunately, we were not able to run these preconditioners on a GPU based library in time and we suppose that an optimized GPU implementation indeed would outperform Metamizer. Thus, we’ll be more careful with our framing of Metamizer and focus more on its versatility with regards to non-symmetric / nonlinear PDEs in our revised manuscript.
> > > > Furthermore, we tested the diagonal Jacobi preconditioner for CG, GMRES and MINRES using the CuPy package (See Section 7.2 in Rebuttal.pdf). Unfortunately, this preconditioner did not help to significantly reduce the number of iterations since the diagonal of the Laplace operator on a regular grid basically corresponds to the identity matrix. However, the preconditioning operations resulted in a slight computational overhead and thus in a slight slowdown of convergence. But we agree that the Jacobi preconditioner could be indeed beneficial for non regular grids and meshes where the diagonal of the Laplace operator exhibits larger variance compared to our regular grids.
> > > >
> > > > We are now doing our very best to update the submission in time and include all of the additional results and changes that were promised to the reviewers.

---

### Author Response · Authors · 2024-11-19

Thanks a lot to all reviewers for their detailed reviews and questions!
We’ll answer all questions and concerns as best we can and look forward to an interesting discussion.
In the meantime, we added extensive baselines to get better quantitative comparisons to traditional numerical linear system solvers (see Rebuttal.pdf in our supplementary).

---

### Author Response · Authors · 2024-11-28

Dear reviewers,

Thanks a lot for your comprehensive feedback, the constructive discussion and suggestions for improvements!
We have done our best to incorporate all your comments and have updated our manuscript as follows:

1. Shortened abstract and shifted focus towards fast and accurate solving of *multiple* PDEs *without retraining*.
2. Improved comparisons to GPU based solvers
3. Additional references, including:
    - “A deep conjugate direction method for iteratively solving linear systems”, Kaneda et al. (2023)
    - “A Neural-Preconditioned Poisson Solver for Mixed Dirichlet and Neumann Boundary Conditions”, Lan et al. (2023)
    - “TENG: Time-Evolving Natural Gradient for Solving PDEs With Deep Neural Nets Toward Machine Precision“, Chen et al. (2024)
    - “Scale-invariant learning by physics inversion”, Holl et al. (2022)
4. Corrected numerous typos and clarified misleading sentences
5. Appendix A: Additional results for turbulent fluids on larger 400x400 domain
6. Appendix B: Domain-size dependent performance comparison for GPU and CPU based solvers
7. Appendix C: Additional comparisons with preconditioned solvers (incomplete LU, algebraic multigrid, Jacobi)
8. Appendix D: Comparison to Newton-CG, nonlinear CG, L-BFGS
9. Appendix E: Comparison of different learning rates for Adagrad, Adam, AdamW
10. Appendix F: Comparison to ground truth of Laplace equation
11. Appendix G: Visualization of gradient norm reduction for cloth simulations
12. Appendix J: Additional details about training pool

A version of our submission with highlighted changes can be found in our supplementary.
Thank you again for helping us enhance the manuscript so much! We hope that this improved version will be of interest to the ICLR audience.

Best regards!

---

### Meta-Review · Area_Chair_DmwM · 2024-12-22

**Metareview:**

**Summary and strengths** The paper presents a learned solver for partial-differential equations (PDEs) which minimizes a physics-based loss function. The method achieves high accuracy on a wide range of PDEs, such as Laplace, advection-diffusion and incompressible Navier-Stokes equation. Unlike other existing neural solvers, at test time, Metamizer can be applied to solve new PDEs not seen during training. such as Poisson, wave and Burgers equation. The approach does not require having a dataset of PDE solutions, as the approach uses the solutions from the previous iterations as the training data.

**Weaknesses** Reviewers pointed out insufficient comparison to the classic solvers like CG, other GPU-based solvers, comparison to other Learning-to-Optimize (L2O) approaches for PDEs. During the rebuttal, the authors added extensive comparisons to address these points.

**Decision** The decision is Acceptance (poster). The paper proposes a simple, yet universal technique that can solve a wide range of PDEs with a single pretrained model, including unseen PDEs, having competitive performance to classic solvers. Reviewers shqZ, EsdF, 8DMR agree that the approach is valuable for the ICLR community and can handle various PDE problems.

**Final version** We suggest the authors to (1) add a specific comparison between a GPU Metamizer to GPU PCG comparison in fig. 17. (2) make more precise claims about the advantages of the method

**Additional Comments On Reviewer Discussion:**

The reviewers had divided views on the paper. In the view of AC, the paper deserves Acceptance as it makes a valuable contribution to neural solvers for PDEs, where a single pretrained network can solve multiple PDEs with competitive performance to classic solvers in both speed and accuracy. Not beating classic, specialized solvers developed over decades of PDE literature is not the grounds for rejection for ICLR.


Summary of most debated points:
- Comparison to other Learning-to-Optimize (L2O) approaches. The authors effectively argued that existing L2O approaches for PDE did not go beyond Poisson Equation and require expensive training data generation, while the proposed approach supports a wide range of PDEs without retraining.
- Lack of comparison to classic solvers and GPU-based solvers. The authors addressed this by adding GPU-based CG, GMRES, MINRES baselines and several other comparisons, demonstrating competitive Metamizer performance in Figures 6 and 14-21.


Summary of the final discussions between reviewers, warranting acceptance:
* Reviewer shqZ strongly advocates for acceptance with score 8
* Reviewer EsdF "does not object acceptance, if CPU to CPU comparisons and add a GPU Metamizer to GPU PCG comparisons will be added in the final paper"
* Reviewer 8DMR "lean towards rejection, but can be OK with acceptance"
* Reviewer gC5k "leans towards rejection, but open to a conditional acceptance if the authors perform a comparison against other L2O methods, the claims are moderated, the abstract is reformatted, and the GPU comparison the other reviewers suggested is performed"

---

### Decision · Program_Chairs · 2025-01-22

Accept (Poster)